# Using wind speed from a blade-mounted flow sensor for power and load assessment on modern wind turbines

Mads M. Pedersen[1], Torben J. Larsen[1], Helge Aa. Madsen[1], Gunner Chr. Larsen[1]

[1] Wind Energy Department, Technical University of Denmark, Frederiksborgvej 399, DK-4000 Roskilde, Denmark

*Correspondence to*: Mads M. Pedersen (mmpe@dtu.dk)

**Abstract.** In this paper an alternative method to evaluate power performance and loads on wind turbines using a blade-mounted flow sensor is investigated. The hypothesis is that the wind speed measured at the blades has a high correlation with the power and loads such that a power or load assessment can be performed from a few hours or days of measurements.

In the present study a blade-mounted five-hole pitot tube is used as the flow sensor as an alternative to the conventional

approach, where the reference wind speed is either measured at a nearby met mast or on the nacelle using LiDAR technology or cup anemometers. From the flow sensor measurements, an accurate estimate of the wind speed at the rotor plane can be obtained. This wind speed is disturbed by the presence of the wind turbine, and it is therefore different from the free-flow wind speed. However, the recorded wind speed has a high correlation with the actual power production as well as the flap-wise loads as it is measured close to the blade where the aerodynamic forces are acting.

Conventional power curves are based on at least 180 hours of 10-min mean values, but using the blade-mounted flow sensor both the observation average time and the overall assessment time can potentially be shortened. The basis for this hypothesis is that the sensor is able to provide more observations with higher accuracy, as the sensor follows the rotation of the rotor and because of the high correlation between the flow at the blades and the power production.

This is the research question addressed in this paper.

The method is first tested using aero-elastic simulations where the dependence of radial position and effect of multiple blade-mounted flow sensors are also investigated. Next the method is evaluated on the basis of full-scale measurements on a pitch-regulated, variable-speed 3.6 MW wind turbine.

It is concluded that the wind speed derived from the blade-mounted flow sensor is highly correlated with the power and flap-wise bending moment, and that the method has advantages over the traditional approach where the met-mast wind speed is

used as reference, e.g. the capability of measuring the shear, veer and turbulence. The aero-elastic simulations show that the assessment time can be reduced, but this reduction cannot be confirmed from the current measurement database due to sensor problems and practical circumstances. Measuring the wind speed at the rotor plane comes with a price as the wind speed is affected by the induction which may be sensitive to the changes you want to evaluate, e.g. different vortex generator configurations. Furthermore it is concluded that a robust instrument and measurement system is required to obtain accurate

and reliable wind speed recordings from pitot-tube measurements.

## 1.1. List of symbols

| | |
|---|---|
| $\alpha$ | Angle of attack |
| $\alpha_p$ | Inflow angle measured by pitot tube |
| $\beta$ | Side slip angle, see Figure 7 |
| $F_\alpha$ | Function mapping $\alpha_p$ into $\alpha$ |
| $F_{vrel}$ | Function mapping $vrel_p$ into $vrel$ |
| $I$ | Inertia of rotor |
| $\omega_{pitch}$ | Angular pitch speed of blade with pitot tube |
| $\omega_{rot}$ | Angular rotational speed |
| $\mathbf{v_{rel}} = \begin{bmatrix} v_{rel,x} \\ v_{rel,y} \\ v_{rel,z} \end{bmatrix}$ | Flow velocity at pitot tube relative to the sensor compensated reduced flow velocity and deflection near the airfoil due to bound circulation |
| $\mathbf{v_{relp}} = \begin{bmatrix} v_{relp,x} \\ v_{relp,y} \\ v_{relp,z} \end{bmatrix}$ | Relative flow velocity measured by the pitot tube including effects of bound circulation |
| $v_{rot}$ | Velocity caused by blade rotation |
| $v_{pitch}$ | Velocity of pitot tube due to pitch motion |
| $\mathbf{v} = \begin{bmatrix} v_x \\ v_y \\ v_z \end{bmatrix}$ | Wind velocity at position of pitot tube including induction effects ($v_y$ is horizontal in direction of main shaft) |
| $\theta_{tilt}$ | Tilt angle |
| $\theta_{rotorposition}$ | Azimuthal angle of blade with pitot tube |
| $\theta_{pitot}$ | Angle of pitot tube relative to the centre line of the blade |
| $\theta_{pitch}$ | Pitch angle of blade with pitot tube |
| $v_{rel}$ | Relative flow speed at pitot tube, compensated reduced flow velocity near the airfoil |

## 2.   Introduction

Detailed knowledge about the wind speed and its variations is essential when evaluating the power performance, load levels

5    and noise migration of modern wind turbines as these properties are highly dependent on the incoming wind conditions (Elliott and Cadogan; Larsen et al., 2005; Barlas et al., 2012; Aagaard Madsen, 2014; St. Martin et al., 2016).

Measuring the correct wind speed is a challenge. Often the wind speed is measured by a cup or sonic anemometer at a met mast two to three rotor diameters away, but if the mean wind direction is not exactly towards the wind turbine, the measured wind will not hit the rotor. Even if the wind direction is exactly towards the turbine, the correlation between the measured

10   wind and the wind at the rotor will decrease with the distance, as smaller turbulence structures will change on their way to

the turbine. Using a proper average time, e.g. 10-min mean values, the temporal and spatial discrepancies are somewhat averaged out, and a good correlation between wind speed and power is achievable.

Another option is to measure the wind with an anemometer mounted on the spinner or the nacelle. In this case the spatial and temporal distance is not an issue, but the measured wind speed is distorted by the rotor. In addition the variation over the rotor plane is often significant due to shear, veer and turbulence - especially in complex terrain and wind farms - and this variation is not captured by an anemometer on the spinner or the nacelle.

LiDAR technology is capable of measuring this variation, but in most setups the LiDAR is configured to scan the inflow at some distance upstream, where the temporal and spatial correlation is lower.

A fourth option is to measure the wind with a blade-mounted flow sensor. In this way the correlation between cause and effect is higher, as the wind is measured exactly where it affects the wind turbine and the effects of shear, veer and smaller turbulence structures are also captured.

Over the last 28 years, blade-mounted five-hole pitot tubes have been used in several research projects to characterize the inflow of wind turbines (Aagaard Madsen, 1991; Petersen and Aagaard Madsen, 1997; Aagaard Madsen et al., 2003, 2010b; Pedersen et al., 2015). Five-hole pitot tubes measure the relative flow velocity as well as the flow angle in two perpendicular planes, and from these quantities three-dimensional turbulent wind speeds can be derived.

In 1989, a pitot tube was mounted on a 95kW Tellus turbine, see Figure 1. The turbine was a fixed pitch, constant speed, stall regulated turbine with rather stiff blades, i.e. the angle of attack, measured by the pitot tube is highly correlated with the axial wind speed at the pitot tube.

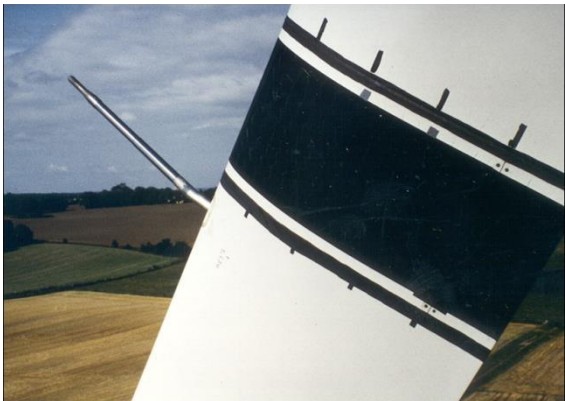

**Figure 1. Five-hole pitot tube mounted on the blade of a 95kW Tellus turbine at Risø in 1995**

A subset of the Tellus measurement dataset has been procured for this study. In Figure 2, the 30-s mean power-production observations are plotted as a function of angle of attack (a) and met-mast wind speed (b). The met mast is located 2.5 diameters from the turbine and observations where the met mast is in wake are excluded from Figure 2 (b). It is seen that the power production correlates much more highly with the angle of attack than with the met-mast wind speed, especially below stall.

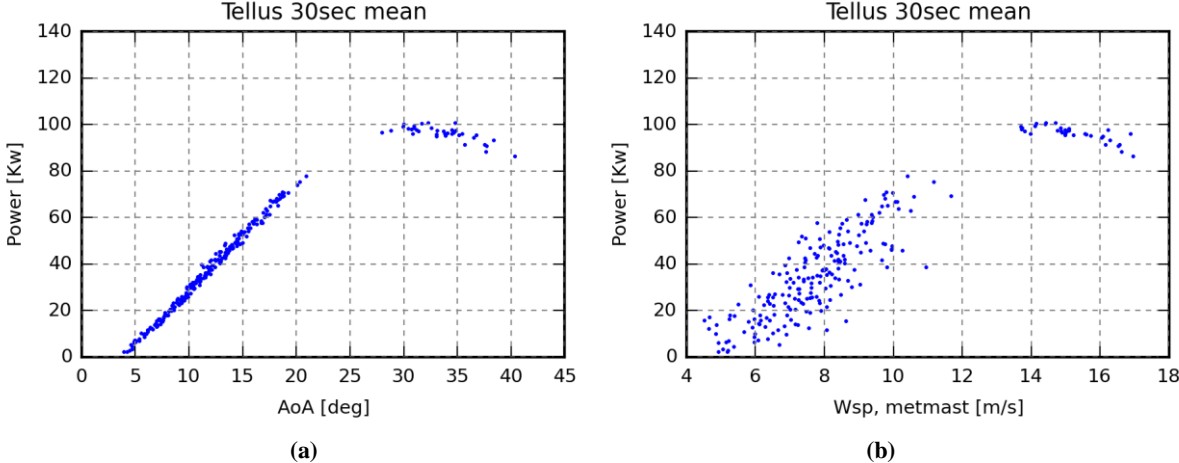

(a)                                                                    (b)

**Figure 2. The 30-s mean electrical power of the Tellus turbine correlates much more highly with the angle of attack (a) than the met-mast wind speed (b)**

The quality of a power curve depends on the number of data points, the scatter of these points and furthermore that all regions of the curve contain enough data points. The number of data points can be increased by extending the measurement period, but it can also be increased by reducing the averaging time. In Figure 3 the averaging time of the pitot-tube-based plot is reduced to the time of one revolution (~1.25 s), i.e. around 24 times more data points are obtained from the same

5     measurement period, and the scatter level is still lower than the met-mast-based 30-s mean observations in the region below stall.

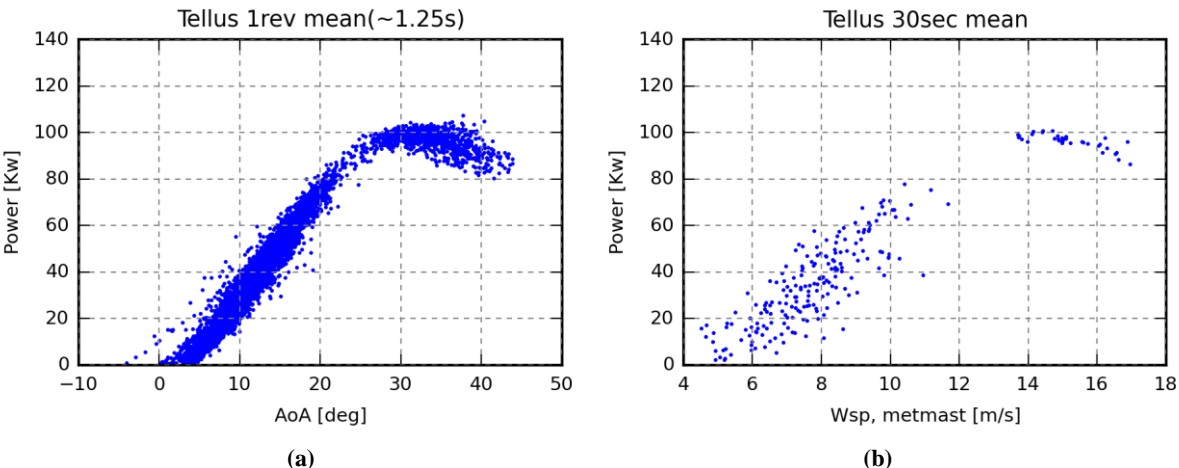

(a)                                                                    (b)

**Figure 3. Despite the much lower average time, one revolution (~1.25 s) instead of 30 s, the average scatter level of the pitot-tube-base observations (a) is still lower than the level of the met-mast-based observations in the region below stall (b)**

This means that the assessment time can be significantly reduced, as many more data points with less scatter are obtained and in addition rare occurring wind speeds are more likely to occur for a 1.25-s than for a 30-s averaging period.

Today, 28 years later, standard wind turbines are pitch regulated, operated with variable speed, have 5-10 times larger rotor

10    and very flexible blades. In this paper we will therefore investigate if a similar speed up in power and flap load assessment time is achievable by using pitot-tube measurements as inflow reference on modern MW wind turbines such that a power curve and load validation can be conducted from a few days of measurements.

The study is based on aero-elastic simulations using the code HAWC2 (Larsen and Hansen, 2007), and measurements on a Siemens 3.6 MW wind turbine.

## 3. Method

In this section the applied procedures for deriving wind speed from pitot-tube measurements are presented as well as the error measure that is used to evaluate the quality of power and flap load curves.

### 3.1. Deriving the wind speed from angle of attack on the Tellus turbine

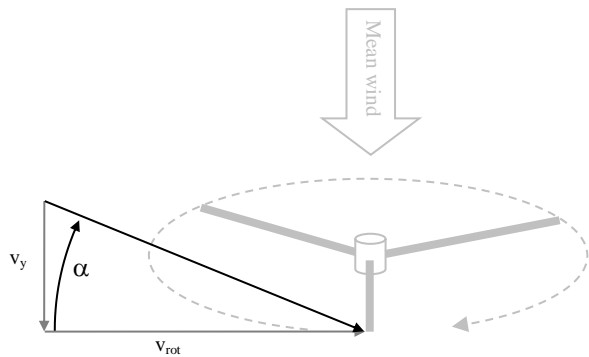

**Figure 4 – In the simple case, the axial wind speed, $v_y$, is a monotonic function of $\alpha$**

For the Tellus turbine which has fixed pitch, constant rotor speed and rather stiff blades, the axial wind speed at the pitot tube, $v_y$, is a function of the rotor speed, $v_{rot}$, and the angle of attack, $\alpha$, see Figure 4:

$$v_y = \tan(\alpha)\, v_{rot} \Rightarrow \alpha \propto \operatorname{atan}(v_y) \tag{1}$$

The inflow angle measured by the pitot tube, $\alpha_p$, obviously depends on the angle of the pitot tube, but also on the position due to increased upwash near the airfoil, see Figure 5. In general the relation between the angle of attack and the flow angle at a point near the airfoil, $F_\alpha$, is nonlinear, but monotonically increasing in the region of interest, if effects of dynamic stall is neglected.

The Tellus turbine has 5° tilt (angle between shaft and horizontal), i.e. $\alpha_p$ is increased when the blade moves up and vice versa, so that $u_p$ becomes:

$$v_y = \cos(\theta_{tilt})\, \tan\!\big(F_\alpha(\alpha_p) + \theta_{pitot} - \theta_{tilt} \sin(\theta_{rotorposition})\big)\, v_{rot} \tag{2}$$

As the sinusoidal contribution from tilt is almost cancelled out when averaging over one revolution, the average axial wind speed can be considered as a monotonic function of $\alpha_p$ when averaging over one or more revolutions.

### 3.2. Deriving the wind speed from pitot-tube measurements of modern wind turbines

For modern wind turbines with variable pitch and rotor speed, $\alpha_p$ cannot be used directly as a measure for the axial wind speed. In this case the following procedure is used:
- Determine the angle of attack and relative velocity from the flow angle and velocity measured by the pitot tube
- Map the angle of attack, relative velocity and pitot-tube side-slip angle from spherical coordinates to 3D cartesian flow vector
- Determine and subtract the velocity due to movement of the pitot tube
- Map the wind speed vector into global coordinates and extract the horizontal component

#### 3.2.1. Estimating angle of attack

At the pitot-tube tip, i.e. near the airfoil, the flow angle, $\alpha_p$, and the relative speed, $|\mathbf{v_{relp}}|$, are different due to local circulation around the blade section and deceleration of the flow, see Figure 5. The first step is therefore to find the angle of attack, $\alpha$, and relative speed, $|\mathbf{v_{rel}}|$.

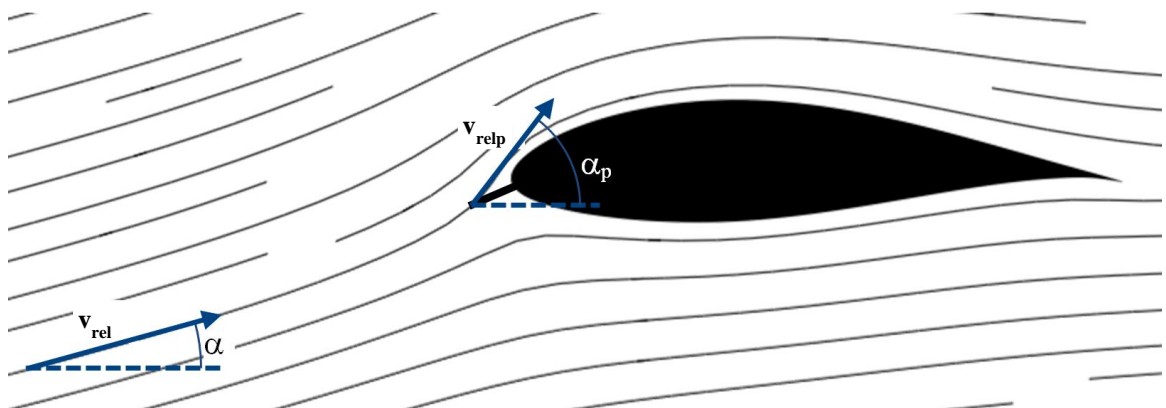

**Figure 5. Map of flow velocities near the airfoil. The flow angle, $\alpha_p$, and velocity, $v_{relp}$, are different due to local circulation and stagnation**

For the simulations this step is not necessary, as HAWC2 directly computes $\alpha$ and $|v_{rel}|$ based on the blade element momentum (BEM) model.

For the measurements, 2D CFD simulations have been used to compute the velocity, $v_{relp}$, for different angles of attack. From these velocities two functions are generated. The first function, $F_\alpha$, maps $\alpha_p$ to $\alpha$, see Figure 6 (a), while the second, $F_{vrel}$, gives the flow speed at the pitot tube, $|v_{relp}|$, relative to the flow speed some distance upstream, $|v_{rel}|$:

$$|v_{rel}| = \frac{|v_{relp}|}{F_{vrel}(\alpha)}$$

[3]

see Figure 6 (b).

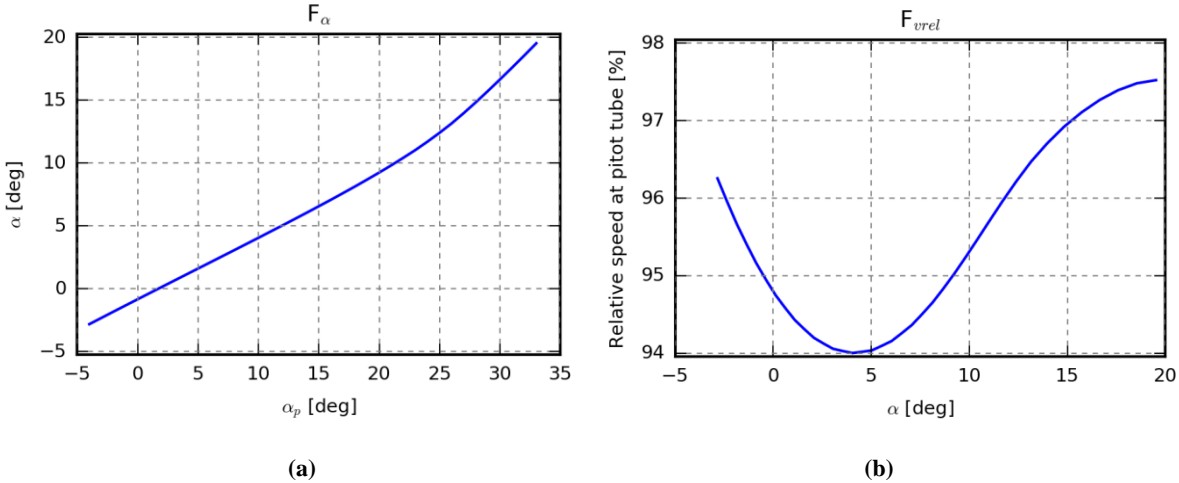

(a)

(b)

**Figure 6. (a) Angle of attack, $\alpha$, as a function of flow angle measured by pitot tube, $\alpha_p$.**
**(b) Relative speed at position of pitot tube as a function of angle of attack.**

### 3.2.2. Map into 3D flow vector
In Figure 7 the relations between the wind speed in polar coordinates, $(\alpha, \beta, |v_{rel}|)$, and Cartesian coordinates $(v_{rel,x}, v_{rel,y}, v_{rel,z})$ are seen.

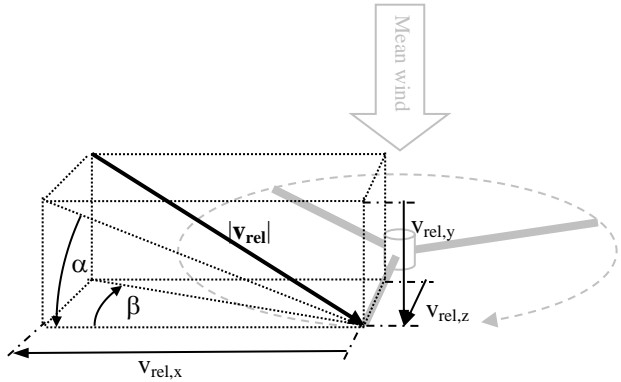

**Figure 7 – Relation between ($\alpha$, $\beta$, $|v_{rel}|$) and $[v_{rel,x}, v_{rel,y}, v_{rel,z}]^T$**

These relations can be formulated as:

$$\tan \alpha = \frac{v_{rel,y}}{v_{rel,x}} \tag{4}$$

$$\tan \beta = \frac{-v_{rel,z}}{v_{rel,x}} \tag{5}$$

$$|\mathbf{v_{rel}}| = \sqrt{v_{rel,x}^2 + v_{rel,y}^2 + v_{rel,z}^2} \tag{6}$$

and now, $\mathbf{v_{rel}} = [v_{rel,x} \quad v_{rel,y} \quad v_{rel,z}]^T$ can be derived:

$$v_{rel,x} = \begin{cases} -\sqrt{\dfrac{|\mathbf{v_{rel}}|^2}{1 + (\tan\alpha)^2 + (\tan\beta)^2}} & ,\text{for } |\beta| \le 90° \\[3ex] \sqrt{\dfrac{|\mathbf{v_{rel}}|^2}{1 + (\tan\alpha)^2 + (\tan\beta)^2}} & ,\text{for } |\beta| > 90° \end{cases} \tag{7}$$

$$v_{rel,y} = \begin{cases} -\sqrt{\dfrac{|\mathbf{v_{rel}}|^2}{\left(\dfrac{1}{\tan\alpha}\right)^2 + 1 + \left(\dfrac{\tan\beta}{\tan\alpha}\right)^2}} & ,\text{for } \alpha > 0 \\[3ex] 0 & ,\text{for } \alpha = 0 \\[3ex] \sqrt{\dfrac{|\mathbf{v_{rel}}|^2}{\left(\dfrac{1}{\tan\alpha}\right)^2 + 1 + \left(\dfrac{\tan\beta}{\tan\alpha}\right)^2}} & ,\text{for } \alpha < 0 \end{cases} \tag{8}$$

$$v_{rel,z} = \begin{cases} \sqrt{\dfrac{|\mathbf{v_{rel}}|^2}{\left(\dfrac{1}{\tan\beta}\right)^2 + \left(\dfrac{\tan\alpha}{\tan\beta}\right)^2 + 1}} & ,\text{for } \beta > 0 \\[3ex] 0 & ,\text{for } \beta = 0 \\[3ex] -\sqrt{\dfrac{|\mathbf{v_{rel}}|^2}{\left(\dfrac{1}{\tan\beta}\right)^2 + \left(\dfrac{\tan\alpha}{\tan\beta}\right)^2 + 1}} & ,\text{for } \beta < 0 \end{cases} \tag{9}$$

### 3.2.3. Estimate and subtract the movement of the pitot tube

The pitot-tube movement derives from three factors; the rotor speed, pitch motions and the speed due to blade deflection.

5 The rotor speed contributes with a tangential speed, $v_{rot}$, which is the product of the rotor speed, $\omega_{rot}$, and the radius of the pitot-tube tip, see Figure 8 (a). Note that the radius changes during pitch motion.

Similarly, pitch motions also result in a velocity, $v_{pitch,}$ tangential to the pitch axis, see Figure 8 (b).

As the speed due to blade deflection cannot be extracted from the current measurement database, this contribution is not included in the present study. The error introduced by this simplification is, however, analysed in Section 4.3.

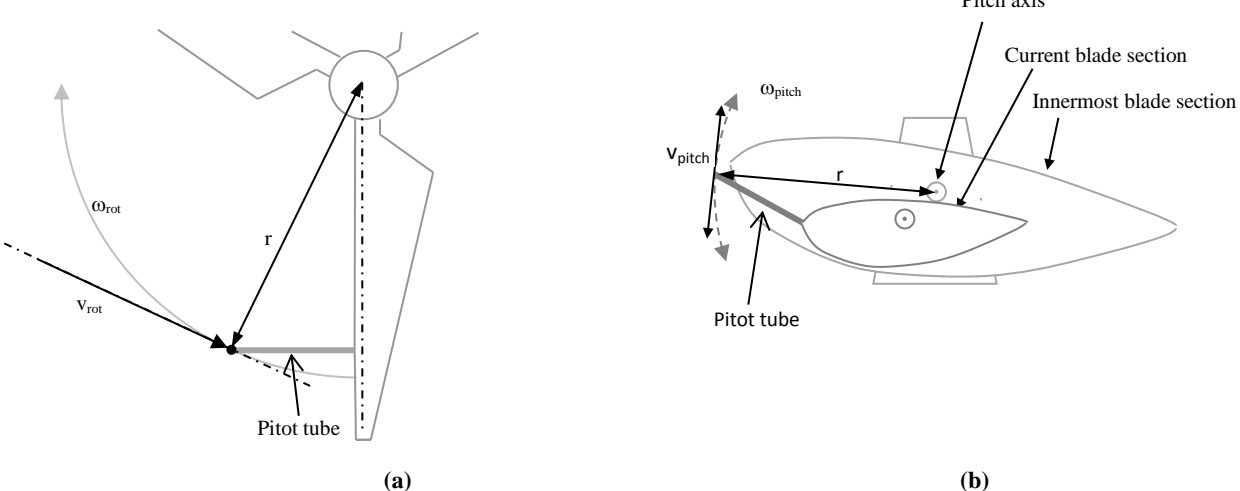

**Figure 8. Rotor rotation (left) and pitch motion (right) moves the pitot tube and contributes to the flow speed measured by the pitot tube**

The measured relative flow velocity, rotational velocity and pitch velocity are now mapped to a common blade coordinate system (indexed B) and subtracted.

$$\mathbf{v^B} = \mathbf{V^B_{rel}} - \mathbf{v^B_{rot}} - \mathbf{v^B_{pitch}} \tag{10}$$

### 3.2.4. Map to global coordinates

Finally the flow velocity is mapped to a global coordinate system using rotation from pitch, coning (downwind angle of blades relative to a line perpendicular to the shaft) , rotor position and tilt, and the horizontal wind speed component parallel to the main shaft, $v_y$, is extracted.

### 3.3. Power of variable-speed wind turbines

Wind turbines convert aerodynamic power to electrical power, but some energy is "stored" as angular momentum in the rotor. When dealing with small time scales on modern variable-speed wind turbines, the fraction of the aerodynamic power used to accelerate or decelerate the rotor may be significant. To include this energy buffer, the power observations used in this study are compensated with respect to rotor speed variations when the power is below rated power, by:

$$\overline{Power} = \overline{Power_{electric}} + \frac{\frac{1}{2}I\left(\omega^2_{rot,t1} - \omega^2_{rot,t2}\right)}{t2 - t1} \tag{11}$$

where I is the inertia of the rotor, $\omega_{rot}$ is the rotational speed, and t1 and t2 are the start and end time of the observation period, e.g. one revolution.

### 3.4. Performance curves

First the observations are binned based on their wind speed values. In this study bins of 0.5 m/s ranging from 3 to 18 m/s are used. For each bin the mean wind speed is calculated as well as the mean power/flap load, and then the power and flap load performance curves are generated by linear interpolation between these mean values.

### 3.5. Error measure

The accuracy of the pitot based power and load curves cannot be evaluated in its present form as the curves cannot be compared to reference curves of existing methods that are based on free flow wind speed. We will therefore focus on variability between periods or cases instead of accuracy.

If the curves obtained in similar conditions are equivalent, then it will be possible to make a small adjustment on the rotor, e.g. changing the tip shape or vortex generator configuration, and determine whether the change affects the power production or load levels.

For this purpose the variation in terms of the mean standard deviation will be used:

For each set of observations, a power or flap load performance curve, PC, is generated. For M different wind speeds the standard deviation of all PCs is then calculated, and the mean standard deviation in percent of maximum power or load is used as an error measure:

$$\text{Variation} = \frac{\frac{1}{M}\sum_{j=1..M}\sqrt{\frac{1}{N}\sum_{i=1..N}\left(PC_i(wsp_j) - \overline{PC(wsp_j)}\right)^2}}{\max(PC)} \qquad [12]$$

In the parameter study referred to in Section 5.2, the effects of different inflow conditions are investigated by changing an inflow parameter, e.g. the turbulence intensity, and comparing the resulting power and load curves to a reference curve. In

this case the error measure is modified, such that the curves are compared to the reference curve:

$$\text{Variation} = \frac{\frac{1}{M}\sum_{j=1..M}\sqrt{\left(PC(wsp_j) - PC_{ref}(wsp_j)\right)^2}}{\max(PC)} \qquad [13]$$

## 4. Numerical study

In this section, simulation results are used to investigate the optimal averaging time, the uncertainties introduced by blade deflection and torsion, the optimal radial position of the blade-mounted flow sensor, and finally a numerically based estimation of the achievable assessment time reduction is presented.

The simulations are performed using HAWC2, which is a nonlinear aero-elastic code intended for computing wind turbine response in the time domain (Aagaard Madsen et al., 2010b, 2012; Kim et al., 2013; Larsen et al., 2015).

The turbine model used for the simulations is based on the structural and aerodynamic configuration of the Siemens 3.6 MW turbine, which was tested at Høvsøre in 2009 during the DANAERO project (Aagaard Madsen et al., 2010a), i.e. a model of the turbine in the full-scale measurement study in Section 5.

**4.1. Simulation overview**

For the analysis, different simulation sets have been created, see Table 1. All simulation sets contain 30 minutes of simulation for each wind speed ranging from 3 to 18 m/s in 0.5 m/s steps, i.e. 15.5 hours of simulation per set.

| Id | No sets | No seeds | Turb. intensity [%] | Shear [power coefficient] | Yaw misalign. [deg] | Air density [kg m⁻³] |
|---|---|---|---|---|---|---|
| SIM1 | 16 | 16 | Meas* | Meas* | 0 | 1.225 |
| Ref | 5 | 5 | 7.5 | 0.1 | 0 | 1.225 |
| Shear | 6 | 1 | 7.5 | 0.01, 0.2, 0.3, 0.4, 0.5, 0.6 | 0 | 1.225 |
| Ti | 7 | 1 | 2.5, 5,.., 20 | 0.1 | 0 | 1.225 |
| Yaw | 5 | 1 | 7.5 | 0.1 | 2.5, 5, 7.5, 10 | 1.225 |
| Dens | 4 | 1 | 7.5 | 0.1 | 0 | 1.175,1.2, 1.25,1.275 |

*For each wind speed the mean turbulence intensity and power shear coefficient are extracted from the measurements

**Table 1. Overview of simulations and parameters**

**4.2. Averaging time**

For IEC standard power curves 10-min mean values are required (IEC 61400-12-1, 2005), but in this study also shorter average times are used.

Reducing the averaging time results in more observations with more variation. If the correlation between wind speed and power/load is high, this variation adds usable information, and the uncertainty of the power/load curve decreases. If on the

other hand the correlation is low, the variation mainly results in more scatter, and nothing is achieved.

To investigate the optimal averaging time, the mean standard deviation of 16 power/load curves have been calculated for different averaging times, ranging from 1 s to 600 s. The 16 curves are based on the 16 simulation sets in SIM1, see Table 1. Figure 9 shows an example of the 16 power curves generated from 15-s mean values. The mean variation of the met-mast-based power curves is 0.21 % of rated power, while it is 0.09 % for the pitot based curves.

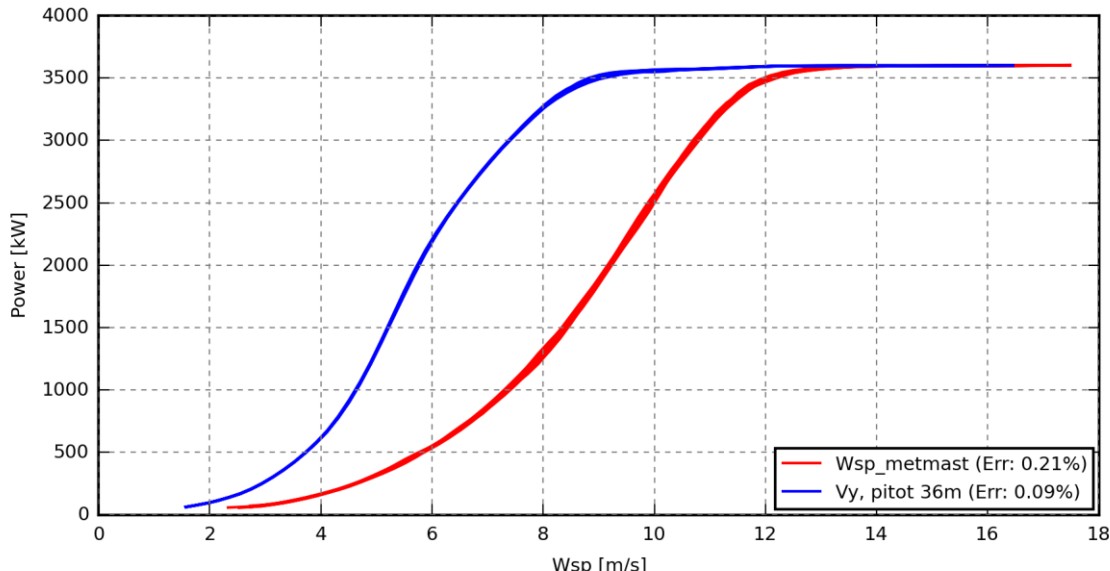

**Figure 9. Example of simulated power curve variation based on 15-s mean values.**
**The variation of the 16 met-mast-based curves is twice the variation of the 16 pitot based curves**

The variation is plotted in Figure 10 as a function of averaging time for power curves based on met-mast wind speeds, free-flow wind speeds at hub centre and wind speeds from a pitot tube at 36 m.

It is seen that the variation is increased with the distance, i.e. the power curves based on met-mast wind speed have more variation than those based on the wind speed at hub centre and the pitot wind speed curves have even less variation. Note

10    that in this case the free-flow wind speed at hub centre is extracted directly from the simulations. This wind speed can be derived from a spinner anemometer or from a nacelle mounted cup anemometer, but in practice additional variation is expected to be introduced by tower deflections and the disturbed-to-free-flow transfer function.

The variation of the hub centre and pitot based curves decreases with shorter averaging times, while the variation of the met-mast-based curves are almost constant due to the lower correlation. In simulations, where the inflow is stationary, the met-

15    mast and hub-centre variations will coincide for longer average times, but in practice long average times are not appropriate due to the non-stationary nature of real inflow.

For average times below one revolution, the variation of the pitot based curves increases significantly as shear- and turbulence-induced local wind speed variations within the rotor are not averaged out.

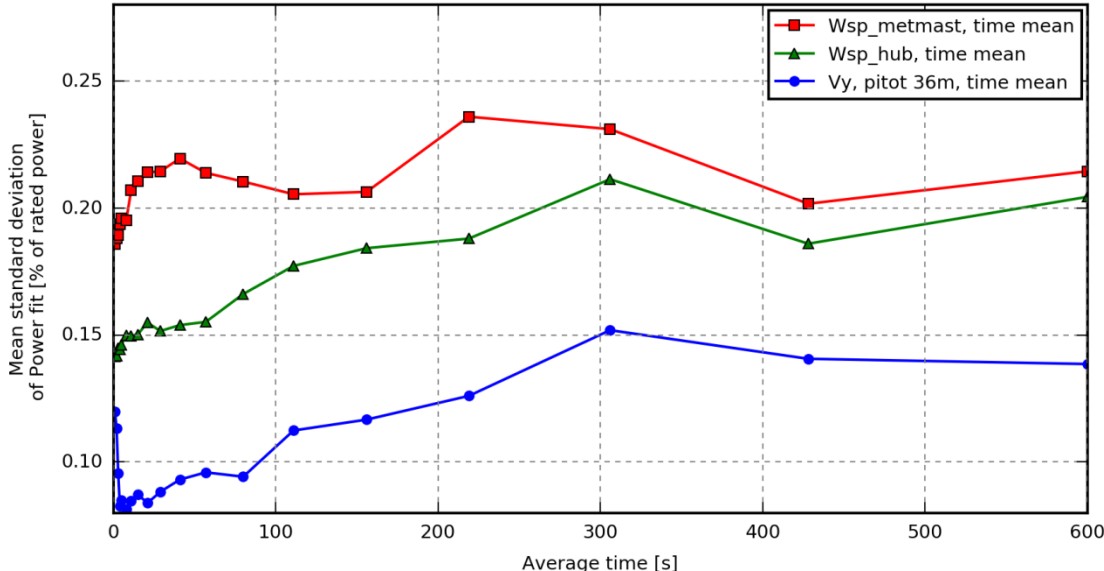

**Figure 10. Mean standard deviation of simulated power curves based on met-mast wind speed, free wind speed at hub centre and wind speed from pitot tube at 36 m. The variation is shown for observation average times ranging from 1 to 600 s.**

Figure 11 shows similar results for the flap moment curves.

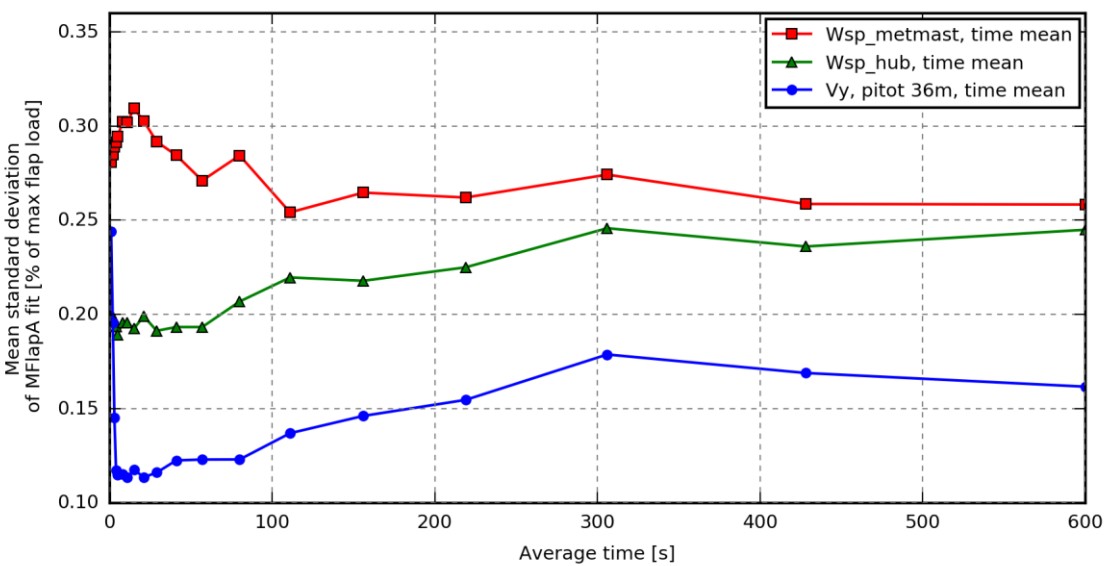

**Figure 11. Mean standard deviation of simulated blade flap moment based on met-mast wind speed, free wind speed at hub centre and wind speed from pitot tube at 36 m. The variation is shown for observation average times ranging from 1 to 600 s.**

### 4.3. Uncertainty due to blade and tower deflection

During operation, the structure of a modern wind turbine moves and the blades deflect and twist considerably. The motion contributes to the movement of the blade-mounted flow sensor and thereby the measured flow velocity. When calculating the wind speed, the velocity of the sensor should be subtracted from the measured flow velocity, but in this case the velocity deriving from blade deflection is unknown and consequently cannot be subtracted. In addition, blade deflection changes the orientation of the sensor such that the measured flow speed cannot be mapped into the true global coordinates, as the true orientation is unknown.

The error that these effects introduce to the estimated wind speed has been investigated using numerical simulations. From the first simulation set in SIM1, see Table 1, pitot-tube data, i.e. $\alpha$, $\beta$, $v_{rel}$, was output at 11 different radial positions on the

blade ranging from the root to the tip. Based on the pitot-tube data, the estimated global wind speed including velocity due to blade deflection was calculated and compared to simulated "real" wind speed.

Figure 12 shows the maximum (a) and mean (b) difference of the axial wind speed component as a function of wind speed and radial position during 30 minutes of operation. As expected the error increases towards the tip, especially in high wind.

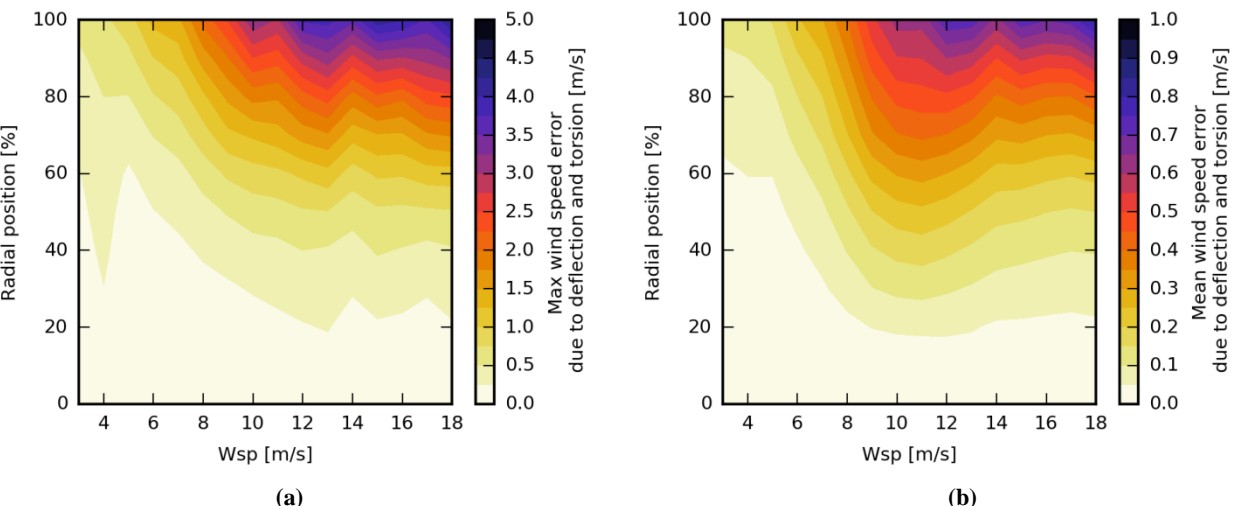

(a)                                                             (b)

**Figure 12. Maximum (a) and mean (b) simulated wind speed error in the axial direction due to deflection and torsion.**

5   At 70 %, i.e. close to the position of the pitot tube in the measurements, the maximum error is around 2 m/s while the mean error is up to 0.5 m/s. This means that the instantaneous wind speed estimated from the pitot-tube data is inappropriate for power and load assessment. In this study however, we use the mean of one revolution as the lowest observation average time, and during one revolution the velocity caused by small scale turbulence induced deflections as well as 1P (one-per-revolution) periodic deflections, e.g. due to shear, is almost averaged out as seen in Figure 13 where the errors of the one-

10  revolution means are shown.

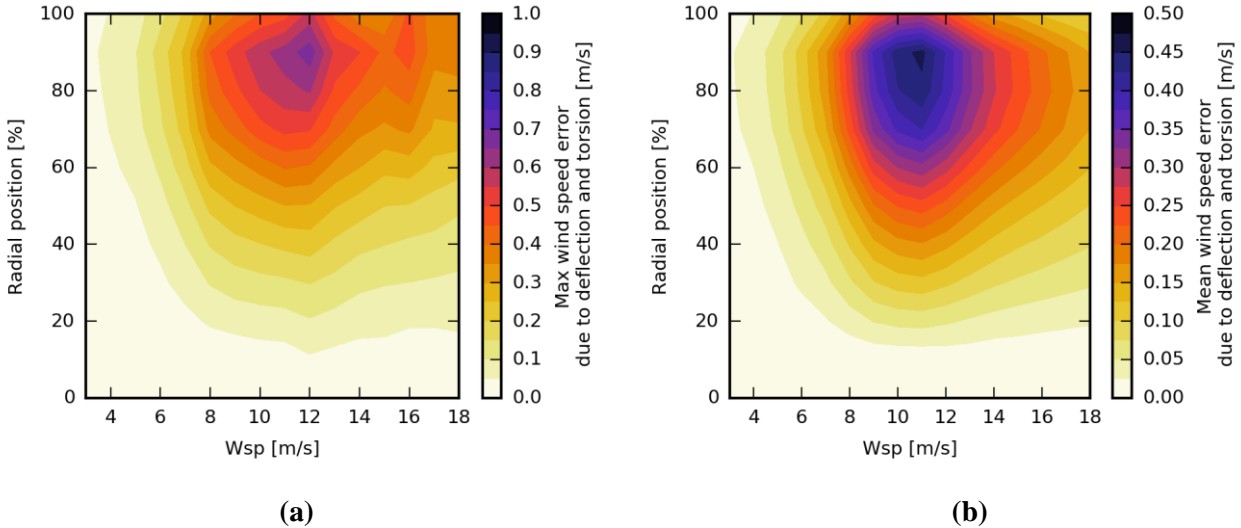

(a)                                                             (b)

**Figure 13. Maximum (a) and mean (b) simulated error of the wind speed (one-revolution mean)**
**in the axial direction due to deflection and torsion.**

The major part of the remaining error is caused by a static deflection of the blade, which changes the orientation of the sensor. This error is related to the thrust on the rotor and peaks around rated wind speed. For a flow sensor mounted at 70 %

in 12 m/s, Figure 14 (a) reveals an almost constant offset between the simulated "real" wind speed and the wind speed estimated from the virtual pitot-tube data, which is confirmed by the error distribution plot in Figure 14 (b).

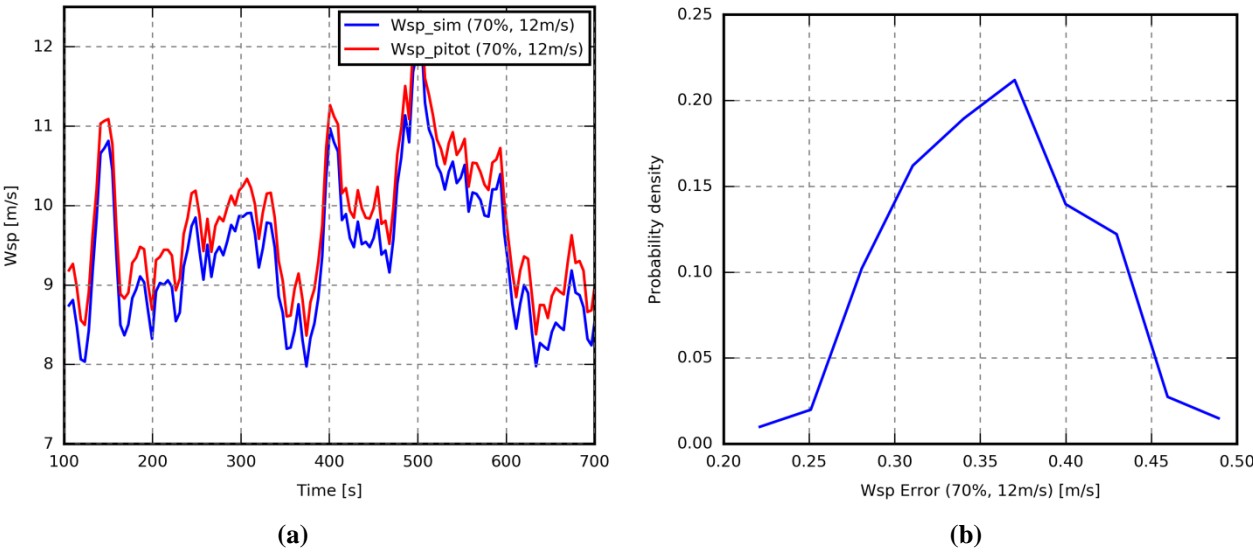

(a)                                                                           (b)

**Figure 14. (a) Time series of simulated "real" wind speed and simulated wind speed estimated from pitot-tube data. (b) Distribution of the simulated error.**

The constant part of the error will shift the x-axis of the power and load assessment curves in a nonlinear way, but it will not introduce additional variation when comparing power and load curves of different periods and it can therefore be neglected in this study.

Figure 15 shows the maximum and mean error of the one-revolution mean wind speed after subtracting the mean offset. At 70 % radial position the maximum and mean error are less than 0.2 m/s and 0.05 m/s respectively.

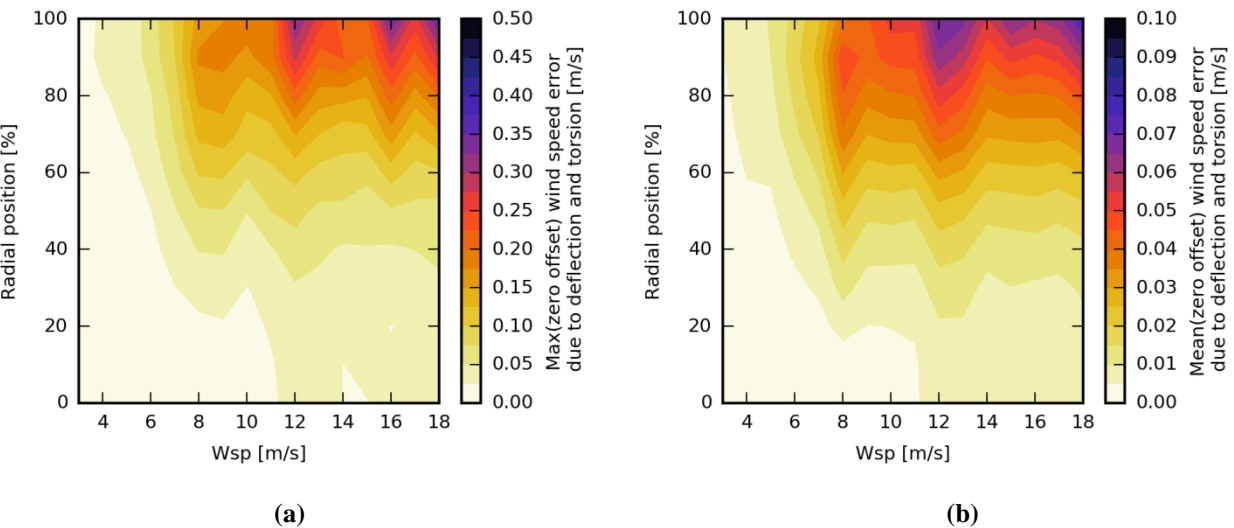

(a)                                                                           (b)

**Figure 15. Maximum (a) and mean (b) simulated error of the wind speed (one-revolution mean neglecting the average offset) in the axial direction due to deflection and torsion.**

### 4.4. Optimal radial position of blade-mounted flow sensors

A flow sensor mounted near the root will only sweep a small area of the rotor plane, while a flow sensor mounted near the tip will suffer from large deflections, torsion and tip loss effects. From the 16 simulation sets in SIM 1, see Table 1, power curves are generated based on one-revolution mean values from pitot tubes at different radial positions. The mean standard deviation of these curves is seen in Figure 16 (a) as a function of radial position. It is seen that a pitot tube at 70 % results in

the lowest variation. This means that the position of the pitot tube in the measurements is close to optimal. Using the mean of the wind speeds measured by the pitot tubes at 20 %, 50 % and 80 % gives power curves with only slightly lower variation. For the flap moment curves, the optimal sensor position is around 50 %, Figure 16 (b). In this case a slightly lower variation is also seen when using the mean wind speed measured by three pitot tubes.

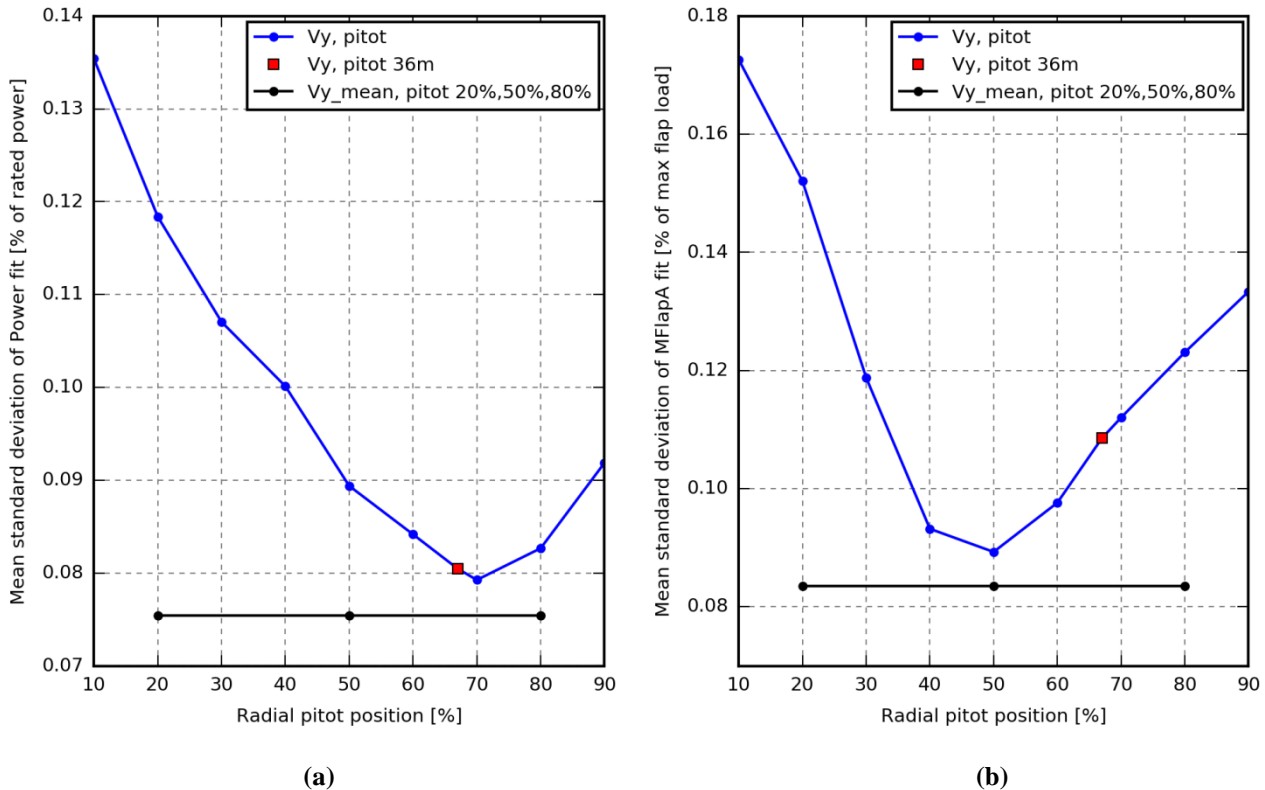

(a)  (b)

**Figure 16. Mean relative standard deviation of simulated power (a) and flap load (b) fits based on wind speed from pitot tubes at different radial positions. The pitot tube at 36 m, which corresponds to the position of the pitot tube in the measurements, is close to optimal for the power curves while the mean of three pitot tubes at 20 %, 50 % and 80 % provides slightly less variation**

### 4.5. Potential assessment time reduction

In this section we want to quantify the potential assessment time reduction that can be achieved - based on simulations. Due to the higher correlation between the blade-mounted flow sensor wind speed and the power/flap loads, it is expected that fewer observations are required to reach a certain variation level. To quantify the reduction, the variance of power curves
10  generated from 1 to 36 hours of one-revolution mean observations is calculated, see Figure 17. The observations used for the power curves are uniformly distributed, i.e. each of the 1 hour power curves is based on approximately 2 minutes of each simulated wind speed from 3 to 18 m/s.

It is seen that it requires 36 hours of observations based on met-mast wind speed and 15 hours of observations based on nacelle wind speed to reach the error level obtained from 4 hours of observations using the pitot tube at 36 m. In general the
15  speed up achieved by using pitot-tube-based observations instead of met-mast-based observations is around 7.

Obviously, the wind speed in real measurement observations is not uniformly distributed - it is not even likely that all wind speeds are observed in the same hour - and therefore a similar speed up cannot be expected in reality.

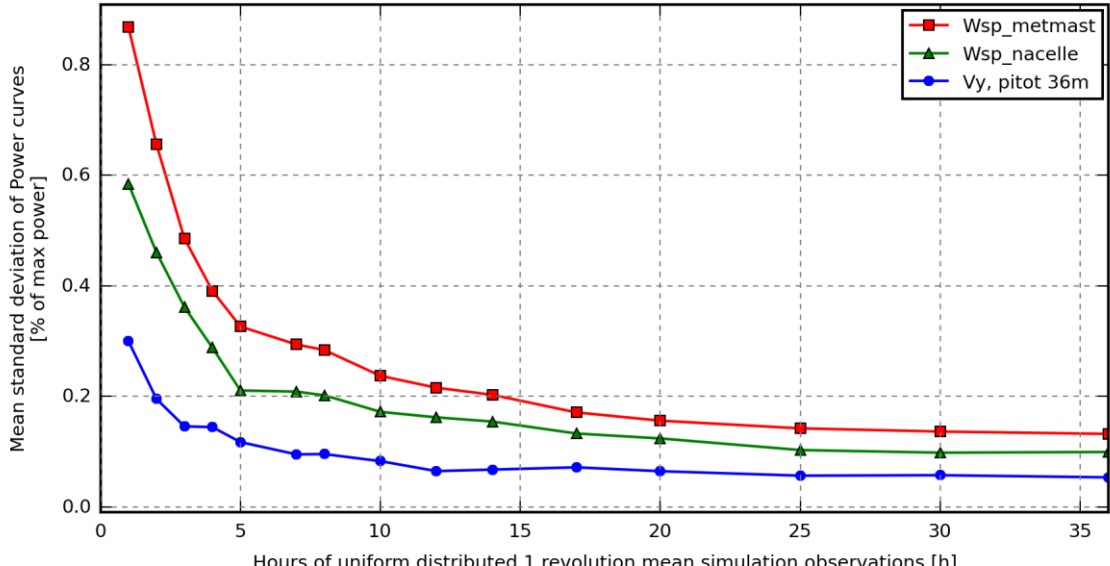

**Figure 17. Mean relative standard deviation of simulated power curves based on uniformly distributed observations. Using the pitot-tube instead of the met-mast wind speed as the basis for the power curves, the same error level can be achieved approximately 7 times faster.**

## 5. Full-scale measurement study

In this section, we investigate whether the wind speeds that can be derived from the pitot tube are also highly correlated with the power and flap loads in practice. The results are based on measurements of a full-scale modern wind turbine with a blade-mounted five-hole pitot tube. Furthermore, we address whether more power and flap moment curves with less

5  variation can be obtained by using pitot-tube wind speed instead of met-mast-recorded wind speeds.

### 5.1. Turbine and site

The measurement data of the 3.6 MW Siemens turbine stems from the DANAERO project 2007-2009 (Aagaard Madsen et al., 2010a). The turbine was located at the Høvsøre test site for large wind turbines in Denmark and equipped with a five-hole CPSPY5 Aeroprobe pitot tube at radius 36 m of one of its 53.5 m blades.

10  The turbine is located in the middle of a row of five large turbines and around three months of measurement data are available on the turbine and the nearby met masts, see Figure 18.

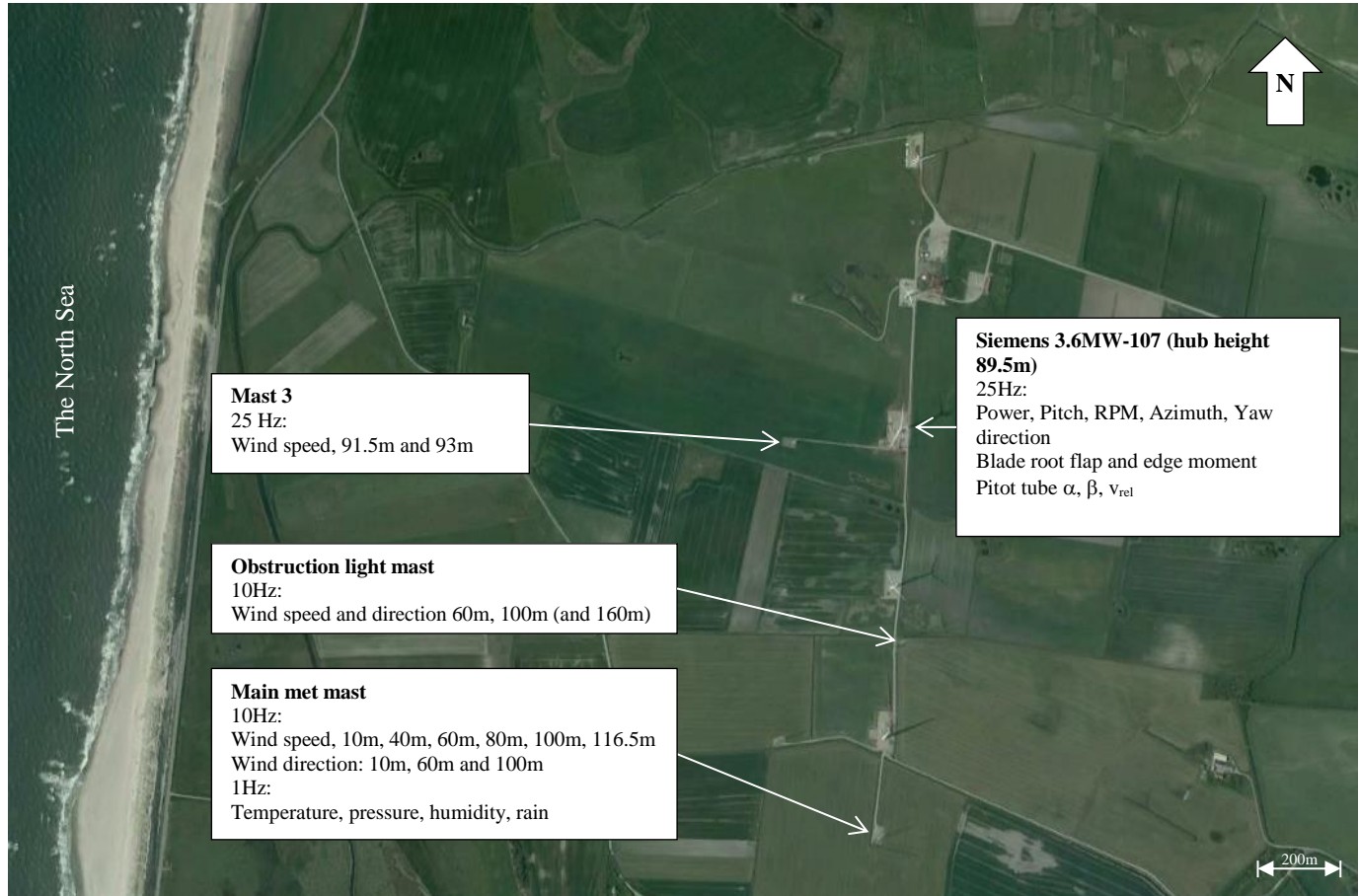

**Figure 18. Overview of the Høvsøre test site for large turbines in Denmark.**
**The Siemens turbine is located in the middle of a row of five turbines.**

## 5.2. Effects of shear and turbulence intensity, yaw error and air density on power and flap load curves

Different turbulence intensity, shear level, yaw misalignment and air density result in different power and load curves (Elliott and Cadogan; Eggers A. J. et al., 2003; Antoniou et al., 2007; Wagner et al., 2011; Fleming et al., 2014; St. Martin et al., 2016). Ideally all observations should therefore have the same turbulence intensity, shear level, yaw misalignment and

air density when calculating the variance of different power and load curves. This is not the case in the measurement dataset and therefore limits must be defined which include observations within a range of turbulence intensities, shear levels, yaw errors and air densities, without changing the power and load curves too much.

These limits have been found by comparing the reference simulation sets with the sets where different turbulence intensities, shear levels, yaw misalignment angles and air densities have been simulated, see Table 1, in terms of the mean absolute

difference between the power and load curves in percent of maximum power and load, cf. eq. [13].

The power and load curves are highly dependent on the turbulence intensity which must be in the range 3−12 % to keep the mean absolute difference below 0.5 %.

For every 10 minutes of measurements, a shear profile coefficient is calculated by fitting a power shear profile to the one hour mean wind speed measured by the Main met mast at 10 m, 40 m, 60 m, 80 m, 100 m and 116.5 m height. Most

observations have a power shear coefficient below 0.3, which has been found to keep the mean absolute difference below 0.5 %, and the rest is discarded.

There is no indication that the yaw misalignment exceeds ±10° and in this region the mean absolute difference of the power and flap moments curves are below 0.5 %.

The aerodynamic power and loads are proportional to the density of air for constant power and load coefficient and therefore

the normalisation

$$\text{Power}_{\text{normalised}} = \text{Power}_{10\min} \frac{\rho_0}{\rho_{10\min}} \qquad [14]$$

must be used for power curves of stall regulated turbines (IEC 61400-12-1, 2005). Pitch-regulated turbines on the other hand reduce the power coefficient at high wind speeds to maintain rated power. Therefore another normalisation, which normalises the free-flow wind speed, is used for pitch-regulated turbines. However in this context we are using the pitot wind speed, and therefore the above normalisation is applied up to rated wind speed from where it is faded out. When using this normalisation no limits are required on the air density to keep the mean absolute difference below 0.5 %.

## 5.3. Selecting measurement observations

A huge effort has been invested in selecting a proper set of observations from the measurement database, as well as in correcting or discarding error prone sensor observations.

The measurement database consists of 1600 hours of measurement, but only a few hundred hours are usable for this analysis, see summary of the selection process in Table 2. The wake filter discards more met-mast observations than pitot observations as the met mast, i.e. Mast3 in Figure 18, is in wake of the turbines in easterly wind directions, but many of these cases are discarded anyway as the turbulence intensity or shear coefficient is outside the accepted range. The final filter that rejects datasets containing corrupted or suspicious pitot observations reduces the amount of observations for the pitot based analysis far below the amount of observations usable for met-mast-based analysis.

| Filter | Pitot observations [hours] | Met mast observations [hours] | Comments |
|---|---|---|---|
| Total | 1599 | 1599 | |
| Normal operation | 926 | 926 | Filter on turbine status signal, rpm>2 and power>100kW |
| Variable rpm | 858 | 858 | In the discarded period the turbine was operated with constant rotor speed |
| No wake | 629 | 528 | Avoid wake situations |
| Turbulence intensity | 537 | 414 | Discard observations with turbulence intensity outside the range 0.03-0.12 |
| Shear | 481 | 414 | Discard observations with power shear coefficient outside the range 0-0.3 |
| Pitot ok | 242 | 407 | Discard observations where pitot-tube data is uncertain, see Section 5.4 |

**Table 2. Measurement selection filters**

## 5.4. Pitot inclusion criteria

The current pitot-tube system is very sensitive to rain, especially the P1-P6 sensor, which records the pressure difference between the centre hole and the static ring. Figure 19 shows the typical behaviour of the P1-P6 sensor, before, during and after rainfall. Before the rain starts, the P1-P6 sensor output is independent of rotor position, Figure 19 (a), and the first droplets are clearly seen as spikes and sinusoidal patterns, Figures 19 (b) and (c). During the rainfall, water gets inside the tubes and the output becomes continuous sinusoidal, Figure 19 (d). During the next couple of hours different more or less abnormal patterns are recorded until the water suddenly disappears and a signal independent of rotor position is restored, Figure 19 (e)-(h). These effects of rainfall have not been detected in the measurements from a previous experiment (Aagaard Madsen et al., 2003), where a Rosemount M858 pitot tube (Schmidt Paulsen, 1990) was used.

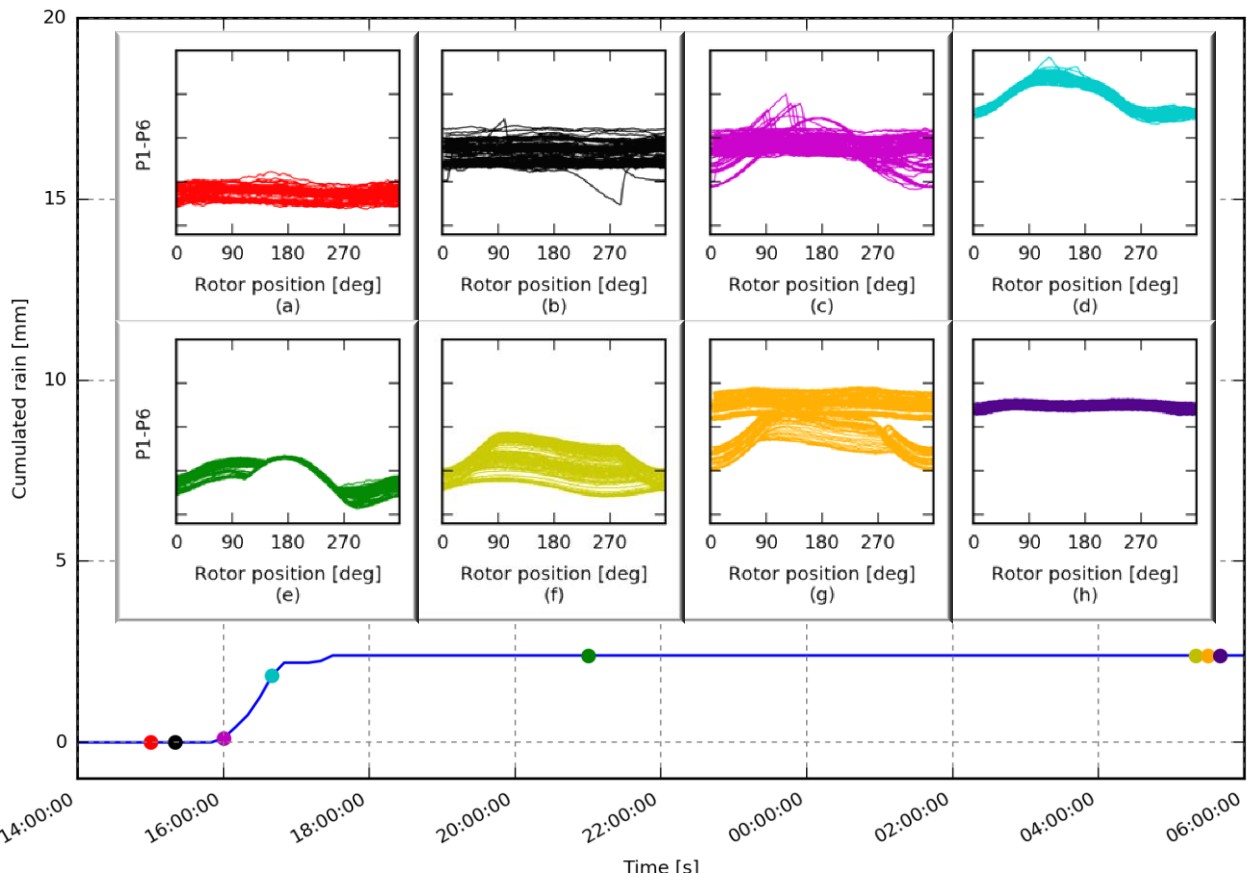

**Figure 19. The background plot shows the cumulated rain from 27<sup>th</sup> - 28<sup>th</sup> April 2009. Approximately 2.5 mm rain fell between 16:00 and 17:30. The coloured points appoint the temporal position of the corresponding graphs on the timeline. These graphs show the P1-P6 sensor (pressure difference between centre hole and static ring) as a function of rotor position (0°: blade down, 180° blade up) over a period of 10 minutes.**
**(a) Before the rain, P1-P6 is almost constant. (b) and (c) First droplets cause spikes and sinusoidal patterns. (c) Water in the tubes is now constantly causing sinusoidal output. (e) Sinusoidal output when the blade is in the upper part only. (f) - (h) Transition from disturbed to normal recording.**

Many of these patterns are clearly abnormal and easy to detect while others are similar to patterns occurring in special inflow conditions, e.g. shear, veer and/or yaw misalignment. It is therefore difficult to make an algorithm that isolates the rain disturbed observations only, and consequently a rough filter that discards observations made between 1 hour prior to rainfall and 12 hours after rainfall is applied in this study.

5   Over an interval of approximately one month in the middle of the measurement period, spikes are often seen in the output of the P1-P6 sensor, see Figure 20 (a). Often the spikes occur at a fixed rotor position for some time, as seen in Figure 20 (b), but the position moves from time to time. In many cases the duration of the spike is only a few degrees as in Figure 20 (b) while in other cases the sensor output level seems to be increased for half a revolution.

A rough filter that discards datasets where the maximum absolute instant change of the P1-P6 sensor output exceeds 50 Pa
10   has been applied.

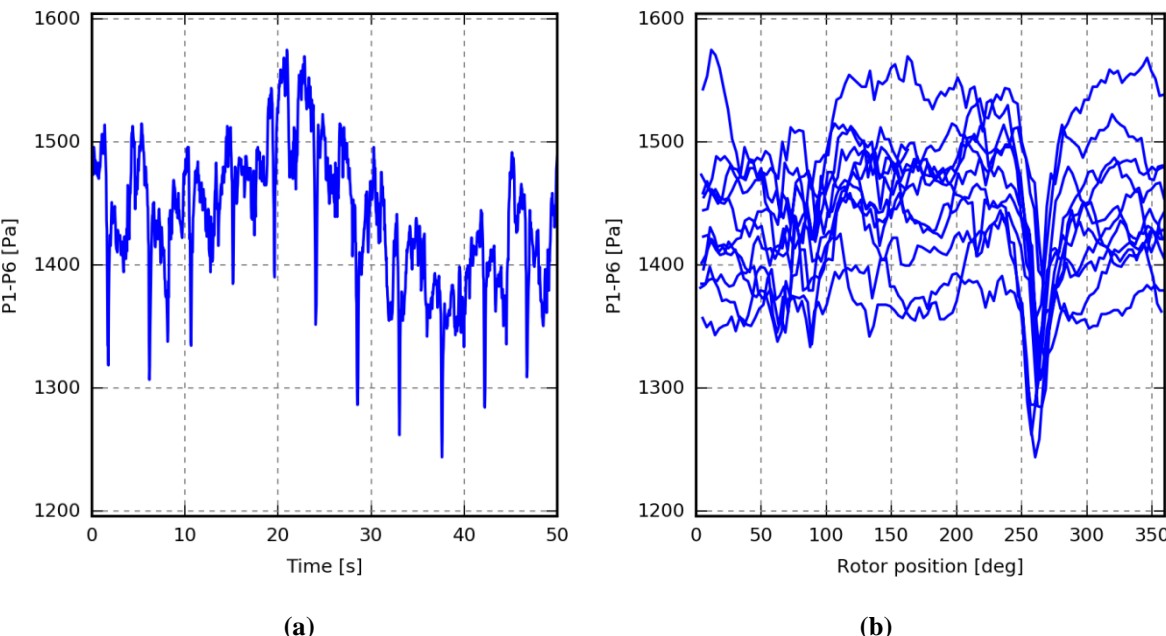

| (a) | (b) |

**Figure 20. Output of the P1-P6 sensor (pressure difference between centre hole and static ring) as a function of time (a) and rotor position (b). Spikes are clearly seen around 260°**

The sensors P2-P3 and P4-P5, which record the inflow and sideslip angle respectively, also have periods with abnormal sinusoidal patterns. These periods are more distinct, but not related to rainfall and in this study they are identified based on their 1P (one-per-revolution) Fourier coefficients values.

### 5.5. Pitot wind speed correlation with power and flap load

5    Figure 21 shows the power, (a) and (c), and flap load, (b) and (d), observations measured on the 9[th] and 10[th] May as a function of pitot and met-mast wind speeds. In (c) and (d) the met-mast observations represent 600-s mean values while all other observations are 120-s mean values. The pitot based observations have less scatter than the 120-s met-mast observations, (a) and (b), while they are comparable to the 600-s met-mast observations which comprise far fewer observations (c) and (d).

10    These results are similar to the results from the Tellus experiment, see Figure 2 and Figure 3, and indicate that it is possible to reduce the scatter of the power and load observations or increase the number of observations by using pitot-tube instead of met-mast recordings - also on modern wind turbines.

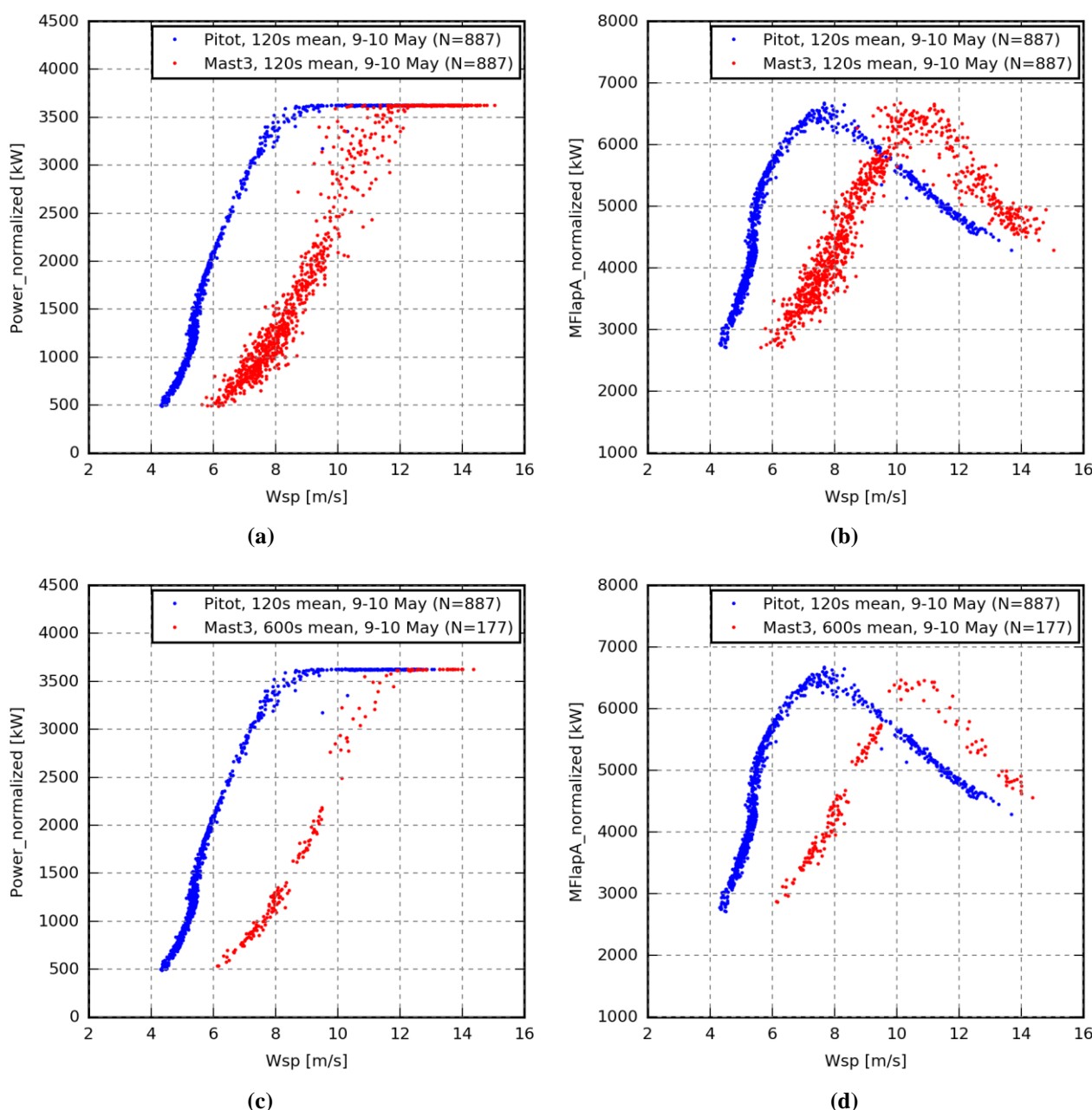

**Figure 21. Measured power (a, c) and flap load (b, d) observations as a function of pitot and met-mast wind speed. In (a, b) all observations are 120-s mean values, while the met-mast observations in (c, d) are 600-s mean values. Using pitot instead of met-mast wind speed, the amount of scatter can be reduced (a, b) or more observations can be obtained (c, d).**

In Figure 21, the pitot observations are based on 120-s mean values. The simulation results in Figure 10, however, indicate that the best results are obtained from averaging times between one revolution and 30 s. Figure 22 (a) shows the power observations from a 10-min period based on 600-s, 120-s and one-revolution mean values. The five 120-s mean values provide information about the slope of the curve in contrast to the single 600-s observation, while the one-revolution mean observations provide information about an even wider range of wind speeds. The scatter of the one-revolution mean observations, however, is remarkable. The scatter is mainly caused by a pitch motion procedure, which changes the pitch angle by one degree every minute to exercise the pitch bearings. These pitch steps result in two levels and increased scatter in both horizontal and vertical directions as both power and pitot wind speeds are affected (Hansen et al., 2005). In Figure 22 (b) the one-revolution mean observations are coloured according to the current pitch state, which is seen to split the observations in two less scattered groups. Unfortunately this means that the averaging time must be at least 120 s to average out the scatter caused by the pitch motion procedure.

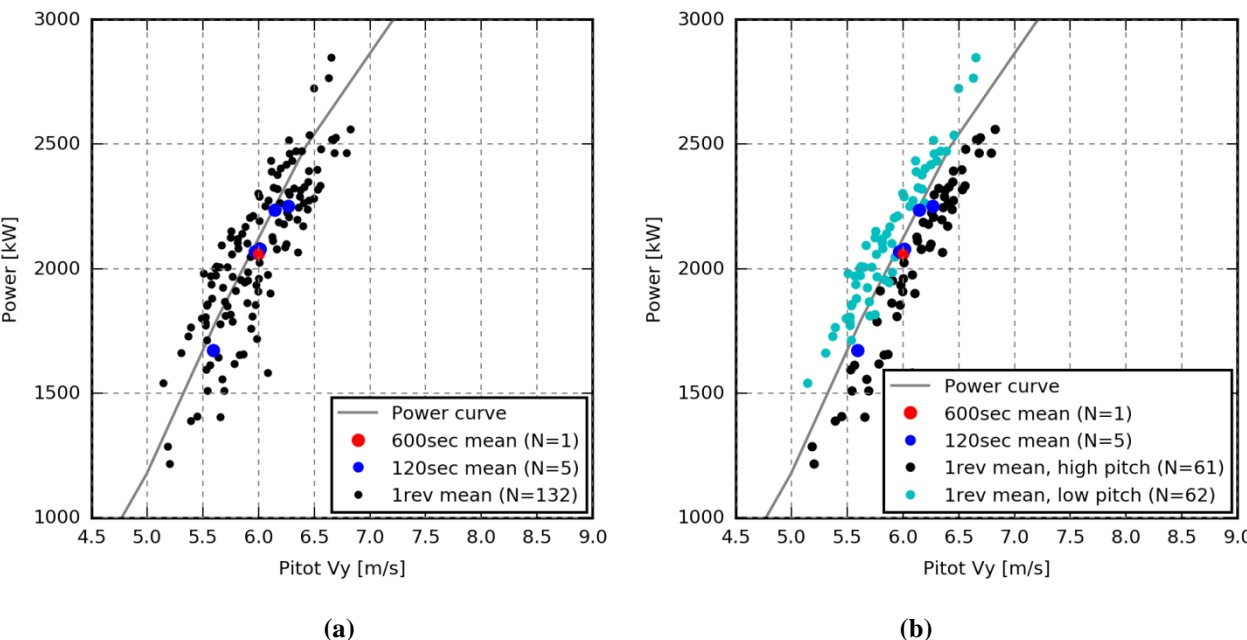

(a)                                    (b)

**Figure 22. (a) Reducing the averaging time increases the number of observations and gives information about a wider wind speed range. Averaging times below 120 s, however, results in a remarkable amount of scatter, which is mainly caused by a pitch motion procedure. (b) The one-revolution mean values are coloured according to the pitch stat, i.e. black points are observations where the pitch angle is increased by 0.5° and cyan points are observations where the pitch angle is decreased by 0.5°. The pitch state splits the observations into two less scattered groups**

In Figure 21, the pitot based power and flap load observations over a two day period were located on a relatively thin line. Plotting all observations, however, results in a thick belt instead of a thin line, see Figure 23 and Figure 24. Colouring the observations according to the time of recording reveals that in many periods the observations are describing a relatively thin line, but the line shifts from time to time.

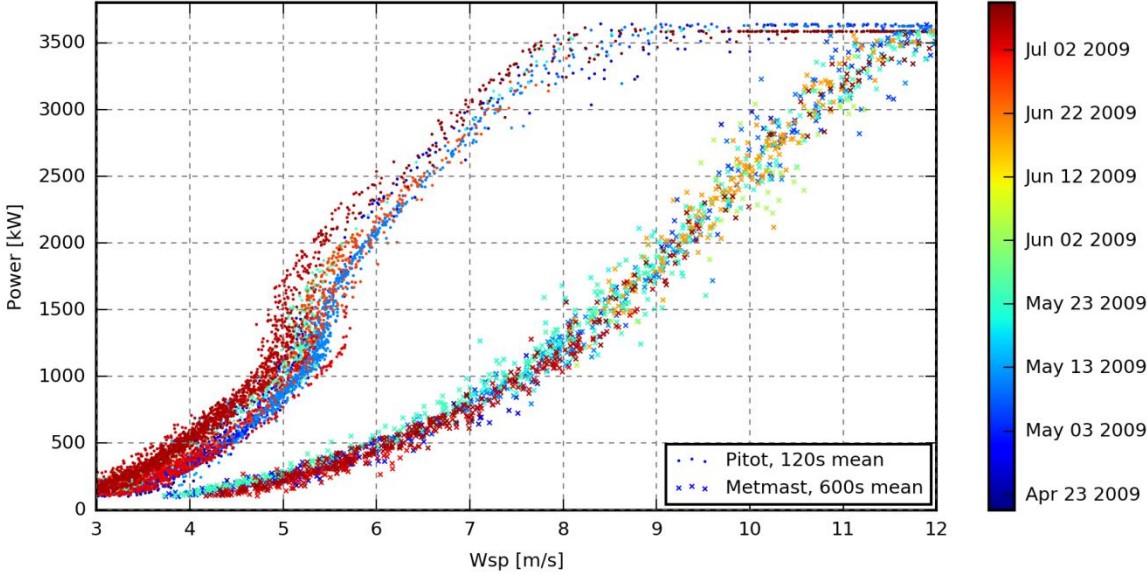

**Figure 23. Measured power observations from the entire database. In many periods, the pitot observations describe a relatively thin line, but the line shifts from period to period**

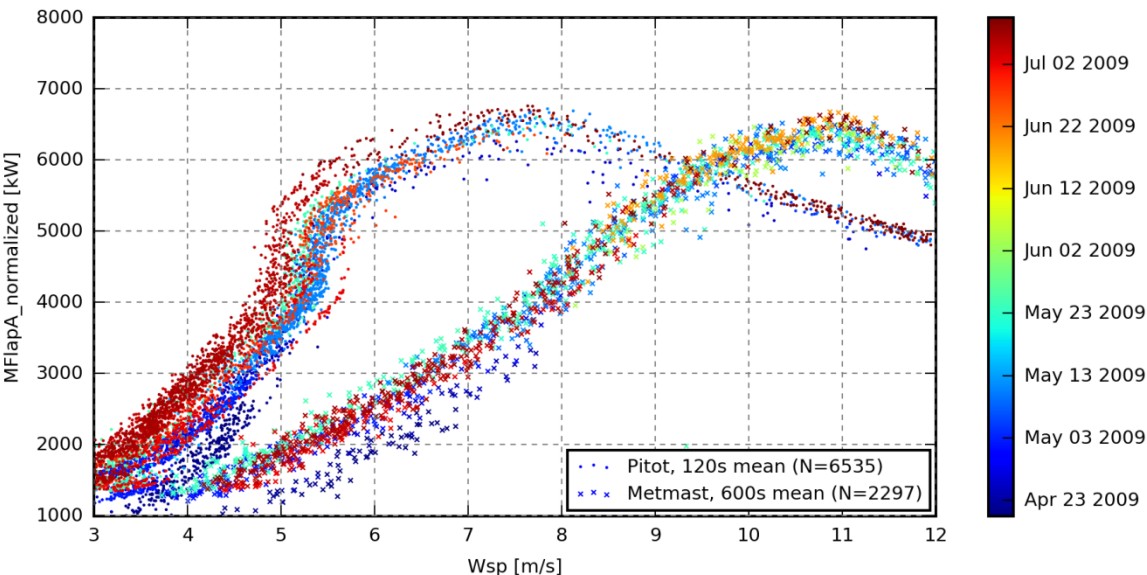

**Figure 24. Measured flap load observations from the entire database. In many periods, the pitot observations describe a relatively thin line, but the line shifts from period to period**

The reason for these shifts has been investigated by comparing time series with similar pitot-tube wind speed and different power levels. In most of these cases the met-mast wind speeds are different, but the pitot-tube wind speeds become equal due to higher loading and thereby different induction. In other cases a combination of fluctuating wind speeds and the pitch
motion procedure results in a different response to similar mean wind speeds. Finally the rated power level clearly changes over the period, see Figure 23. This indicates that the control settings are not fixed and it is therefore likely that also the behaviour at lower wind speeds changes during the measurement period.

### 5.6. Number and variation of performance curves

In Section 4, the comparison of uniformly distributed simulation observations showed that the number of hours required to
obtain power curves with a variation level below a certain threshold can be reduced around 7 times by using pitot based instead of met-mast-based wind speeds. This result was based on one-revolution mean values, which in Section 4.2 were found to give the lowest variation. However, this is not feasible for the measurements, due to the scatter introduced by the pitch motion procedure.

In practice it is not possible to obtain a full power curve from e.g. every 4 hours, as both low and high wind situations must
occur, and from the current measurement database only a few power curves can be obtained. The exact number of curves depends on the wind speed range and resolution of interest. In the following analysis, the range from 4 to 13 m/s (met-mast wind speed) is divided into bins of 0.5 m/s. Observations are then collected for the first power curve until all bins contain at least three observations, then for the next power curve etc.

Figure 25 shows the resulting power curves and the underlying observations. In this case the pitot based approach provides
three curves with 1.1 % variation (percent of maximum power) while four curves with slightly higher variation are obtained from the met-mast-based observations. This means that from the current measurement database, an assessment time reduction cannot be achieved by using pitot-tube wind speed instead of met-mast wind speed.

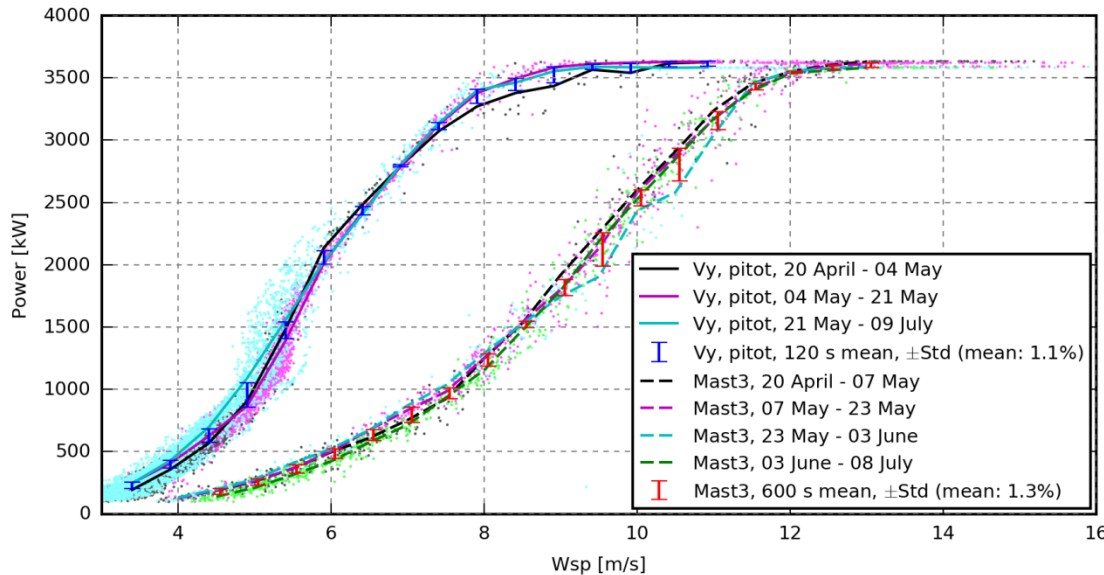

**Figure 25. Only two or three full range power curves can be obtained from the current measurement database.**

This result is highly constrained by the pitch motion procedure that introduces scatter for averaging times below 120 s, as well as by the amount of discarded observations due to abnormal pitot sensor values. These constraints may be different for another turbine and pitot-tube instrument. An indication of the optimal potential has been achieved by relaxing the
constraints, i.e. suppressing the pitot ok filter and using one-revolution mean values. In this optimal case, the pitot based approach results in 20 power curves, and the 19[th] June produces four power curves in a single day, i.e. the same number as obtained from the whole period using the met-mast-based approach. Note, however, that the variation of these curves may be significant due to shorter averaging times and the line shifting seen in Figure 23 and Figure 24.

## 6.   Conclusion

In the numerical part of this paper it is shown that the wind speed derived from a blade-mounted flow sensor of a modern pitch-regulated, variable-speed wind turbine is higher correlates more highly with the power and flap loads than the wind speed measured 2.5 diameters upstream. The correlation is also higher than the wind speed that can be obtained from a fixed point instrument, e.g. a nacelle mounted cup anemometer.

When using wind speed from a blade-mounted flow sensor as the basis for power and flap load curves, shorter observation
average times, in the range 5 - 90 s, reduce the variation of the curves.

Deflection and torsion of the blades introduce an error on the wind speed derived from a blade-mounted flow sensor. When averaging over one or more revolutions, the error is significantly reduced and the major part of the remaining error is a constant offset, which can be ignored when power and flap load curves of different periods are compared. For the current turbine the radial flow sensor position that results in the lowest variation of power curves was found to be 70 %, while a
sensor in 50 % is optimal for flap load curves.

Finally, the analysis showed that the length of the assessment period, which is required to achieve a certain power or flap load curve variation can be reduced around 7 times by using uniformly distributed wind speed observations from a blade-mounted flow sensor instead of a met mast.

The measurement part of the paper concludes that the current pitot-tube system is highly sensitive to rain, and a proper algorithm must be applied to discard error prone observations.

During shorter periods the pitot-tube wind speed often correlates more highly with power and flap loads than the met-mast wind speed, such that the scatter can be reduced or the number of observations increased by reducing the observation average time. During longer periods however, the scatter of the pitot-tube-based power and flap load observations becomes significant and often the observations seem to group around relatively thin lines that shift from period to period.

From the current measurement database, an assessment time reduction cannot be achieved using pitot-tube wind speeds instead of met-mast wind speeds. A limiting factor for the present data set is the pitch motion of 1° every minute that introduces severe scatter when applying averaging times below 120 s and the high number of observations that are discarded to avoid error prone pitot observations. Relaxing these constraints reveals that the method may have a high potential for turbines without the pitch motion procedure and equipped with a more robust flow sensor system.

The wind speeds measured by a blade-mounted flow sensor have advantages over the wind speeds measured at the met mast, as it can be used offshore, in wind farms and in complex terrain, where it may not be possible to put a met mast. It can be used to investigate e.g. aerodynamic modifications or detect performance issues, e.g. due to leading edge roughness by comparing the relative pitot based power and load curve between different periods or turbines. In addition it follows the yaw

direction, i.e. it is never in wake of the current turbine in contrast to a traditional met mast. Moreover additional information about angle of attack, wind speed variations within the rotor plane, e.g. shear, veer, height dependent turbulence intensity and wake effects can be extracted. This information can be used as input for control of individual pitch or active trailing edge flap to optimise power and/or reduce loads or noise (Larsen et al., 2005; Barlas et al., 2012; Kragh and Hansen, 2012; Kragh et al., 2012; Aagaard Madsen, 2014).

The measured wind speeds are however affected by the wind turbine induction. This means that it cannot be used for IEC standard power curves, and changing the induction, e.g. by another control strategy, yaw misalignment or by the pitch step procedure seen in the present study, will shift the resulting wind speed. Another problem is that the induced wind speeds increase up to rated power, i.e. in this region the flow sensor based power curve has a higher slope, which for the present

turbine was found to be almost vertical around rated rotor speed, such that small wind speed uncertainties result in large power variations.

It is possible to compensate for the presence of the turbine using an aerodynamic model (Pedersen et al., 2015), but it requires detailed knowledge about the aerodynamic properties of the blades, several assumptions and compromises, and it adds additional uncertainty.

Finally the wind speeds derived from pitot-tube measurements are more vulnerable to uncertainties and errors in the measurement system, as they are derived from 10 sensors instead of one. Therefore a robust measurement system is required and the pitot-tube instrument and pressure transducers must be designed to prevent rain, moist and dust from entering the tubes by using a heater, or a draining and/or a pneumatic cleaning system etc.

It is therefore concluded that the wind speed measured by a blade-mounted flow sensor is highly correlated with the power and flap loads, especially during shorter periods, but the potential assessment time speed up that was obtainable in the simulation, could not be confirmed from the current measurements.

Blade-mounted flow sensors are, however, able to provide additional valuable information about the inflow variations within the rotor plane, but a robust instrument and measurement system is required to extract reliable wind speed measurements.

**7. Data availability**

Measurement and simulation data not available due to confidentiality.

## 8. Competing interests

The authors declare that they have no conflict of interest.

## 9. Acknowledgement

The authors would like to acknowledge Siemens Wind Power for providing data for the simulation model and the funding
from the Danish Energy Agency EUDP programme of the DANAERO MW projects, contracts ENS no. 33033-0074 and
64009-0258 are acknowledged for providing important data for the present study.

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
