# Peer review of "Using wind speed from a blade-mounted flow sensor for power and load assessment on modern wind turbines"

_Wind Energy Science, 2017_

## Referee Comment (RC1) · Anonymous Referee #1 · 21 Jul 2017

This paper discusses the use of flow sensors on the blades as a means for determining the power curve and loads on the blade. Rigorous mathematical discussion is shown on how to remove interactions within these measurements to achieve a "clean" measurement. The advantage of this approach is shown to be the closeness of the measurement to the turbine, which enables less variability for similar conditions. Thus, the power curve can be extracted at a much quicker rate.

My only major comment on the paper is that I did not see any discussion on the accuracy of the power curve and loads extracted using this approach - only the variability was focused on. And, to get an accurate power curve, you will need to adapt the mea-

surements based on the interactions of the turbine. Will there need to be a turbine-specific adaptation of the measurements to extract a meaningful power curve? Thus necessitating a rigorous analysis of the process for each application?

The paper is well written and understandable, but there are still numerous grammatical errors - too many to correct in this context. A thorough edit of the paper is needed. "Brake" should be "break". Also, not all variables used in the equations are identified.

---

## Referee Comment (RC2) · V.A. Riziotis (Referee) · 1 Aug 2017

The paper presents a method for the assessment of the average power curve and loads curves of a wind turbine using a blade mounted pitot tube. The method is evaluated numerically, on the basis of aeroelastic simulations performed with HAWC2 code on a 3.6 MW turbine, and experimentally using measured data from the same turbine. The method is compared against standard wind speed measurements performed using a met mast situated a few diameters upstream of the rotor but also hub velocity measurements based on a spinner anemometer (the latter was only assessed numerically). In the paper it has been clearly shown that the method presents certain

advantages compared to standard measurement techniques such as reduction of the scatter of the measured data even for lower averaging periods which has an additional advantage that resulting power curves are more populated, including more data points. On the other hand, the authors could not prove that application of their method leads to a reduction of the assessment time as anticipated through their numerical analysis. However, this was only due to a special function of the pitch control of the specific turbine in partial load operation. The paper is well written and presents innovative work on the field of power and loads curves assessment. Also, the conclusions drawn and the advantages and disadvantages of the method are very clearly demonstrated and explained both in the text and in the conclusions section. Two specific comments that could be further discussed in the revised text are given below: 1) The method in its present form cannot provide wind speed measurements due to the induction effect. This is of course acknowledged by the authors in the conclusion section. This means that it cannot be used for power curve certification at least in its present form. In connection to the above the authors could elaborate further a) on the potential application/use of the approach in its present form (e.g. power control) b) they could also discuss any recent developments in the direction of correcting the measured wind speed for axial induction effects. 2) One of the shortcomings of the method is that the sensor follows the deflections of the blade/tower. Since the deflections of the turbine cannot be measured, the above effect cannot be corrected for. The authors numerically analyze the effect of blade deflections on the error of the wind speed measurements and they find that this is thrust driven. In the results of fig. 12 it is seen that the error is almost constant over the whole full load range (especially the maximum error while the mean error seems to be indeed higher around the rated speed). So there seems to be some significant contribution also from blade torsion given that thrust (and flapwise deflections) will be relatively low at 18m/s. It might be interesting to make the distinction of the above two effects in your analysis. Editorial changes/modifications are discussed in the accompanying pdf.

Please also note the supplement to this comment:
https://www.wind-energ-sci-discuss.net/wes-2017-25/wes-2017-25-RC2-
supplement.pdf

**Supplement:**

[revised manuscript text omitted]

A subset of the Tellus measurement dataset has been procured for this study. In Figure 2, the 30s mean power production observations are plotted as function of angle of attack (a) and met mast wind speed (b). The met mast is located 2.5 diameters from the turbine and observations where the met mast is in wake are excluded from Figure 2 (b). It is seen that the the power production is much more correlated with the angle of attack than with the met mast wind speed, especially below

5  stall.

[Figure]

(a)                                                                (b)

**Figure 2. The 30s mean electrical power of the Tellus turbine is much more correlated with the angle of attack (a) than the met mast wind speed (b)**

The quality of a power curve depends on the number of data points, the scatter of these points and furthermore that all regions of the curve contain enough data points. The number of data points can be increased by extending the measurement period, but it can also be increased by reducing the averaging time. In Figure 3 the averaging time of the pitot tube based plot is reduced to the time of one revolution (~1.25 s), i.e. around 24 times more data points are obtained from the same

10  measurement period, and the scatter level is still lower than the met mast based 30 s mean observations in the region below stall.

[Figure]

[Figure]

**Figure 3. Despite the much lower average time, one revolution (~1.25s) instead of 30s, the average scatter level of the pitot tube base observations (a) is still lower than the level of the met mast based observations in the region below stall (b)**

This means that the assessment time can be significantly reduced, as much more data points with less scatter are obtained and in addition rare occurring wind speeds are more likely to occur for a 1.25 s than for a 30 s averaging period.

Today, 28 years later, standard wind turbines are pitch regulated, operated with variable speed, have 5-10 times larger rotor and very flexible blades. In this paper we will therefore investigate if a similar speed up in power and flap load assessment

5   time is achievable by using pitot tube measurements as inflow reference on modern MW wind turbines such that a power curve and load validation can be conducted from a few days of measurements.

The study is based on aero-elastic simulations using the code HAWC2 (Larsen and Hansen, 2007), and measurements on a Siemens 3.6MW wind turbine.

[Figure]

[Figure]

**3. Method**

In this section the applied procedures for deriving wind speed from pitot tube measurements are presented as well as the error measure that is used to evaluate the quality of power and flap load curves.

**3.1. Deriving the wind speed from angle of attack on the Tellus turbine**

[Figure]

**Figure 4 – In the simple case, the axial wind speed, u, is a monotonic function of α.**

5   For the Tellus turbine which has fixed pitch, constant rotor speed and rather stiff blades, the axial wind speed at the pitot tube, $u_p$, is a function of the rotor speed, $v_{rot}$, and the angle of attack, $\alpha$, see Figure 4:

$$u_p = \tan(\alpha)\, v_{rot} \Rightarrow \alpha \propto \mathrm{atan}(u_p) \qquad [1]$$

The inflow angle measured by the pitot tube, $\alpha_p$, obviously depends on the angle of the pitot tube, but also on the position due to increased upwash near the airfoil, see Figure 5. In general the relation between the angle of attack and the flow angle at a point near the airfoil, $f_{\alpha_p \to \alpha}$, is nonlinear, but monotonically increasing in the region of interest, if effects of dynamic

10   stall is neglected.

The Tellus turbine has 5° tilt (angle between shaft and hozizontal), i.e. $\alpha_p$ is increased when the blade moves up and vice versa, and $u_p$ becomes:

$$u_p = \cos(\theta_{tilt}) \tan\left( f_{\alpha_p \to \alpha}(\alpha_p + \theta_{pitot}) - \theta_{tilt} \sin(\theta_{rotorposition}) \right) v_{rot} \qquad [2]$$

As the sinusoidal contribution from tilt is almost cancelled out when averaging over one revolution, the average wind speed, U, can be considered as a monotonic function of $\alpha_p$ when averaging over one or more revolutions.

**3.2. Deriving the wind speed from pitot tube measurements of modern wind turbines**

15   For modern wind turbines with variable pitch and rotor speed, $\alpha_p$ cannot be used directly as a measure for the axial wind speed. In this case the following procedure is used:

- Determine the angle of attack, $\alpha$, and relative velocity, $v_{rel}$, from the flow angle, $\alpha_p$, and velocity, $v_{relp}$, measured by the pitot tube

20   - Map the spherical coordinates $v_{rel}$, $\alpha$, and the pitot tube sideslip angle, $\beta$, into the cartesian 3D flow vector, $\mathbf{V}_p$, see Figure 7
- Determine and subtract the velocity due to movement of the pitot tube
- Map the wind speed vector into global coordinates and extract $u_p$

**3.2.1.   Estimate angle of attack**

[revised manuscript text omitted]

$$\mathbf{W}_{p\_bl} = \mathbf{V}_{p\_bl} - \mathbf{v}_{rot_{bl}} - \mathbf{v}_{pitch\_bl} \qquad [10]$$

**3.2.4.   Map to global coordinates**

Finally the flow velocity, $\mathbf{W}_{p\_bl}$, is mapped to a global coordinate system using rotation from of pitch, coning (downwind

5   angle of blades relative to a line perpendicular to the shaft) , rotor position and tilt, and the wind speed component parallel to
the yaw direction, $u_p$, is extracted.

**3.3. Power of variable-speed wind turbines**

Wind turbines convert aerodynamic power to electrical power, but some energy is "stored" as angular momentum in the
rotor. When dealing with small time scales on modern variable-speed wind turbines, the fraction of the aerodynamic power

10   used to accelerate or decelerate the rotor may be significant. To include this energy buffer, the power observations used in
this study are compensated for rotor speed variations when the power is below rated power, by:

$$\overline{Power} = \overline{Power_{electric}} + \frac{\frac{1}{2}I(\omega_{rot,t1}^2 - \omega_{rot,t2}^2)}{t2 - t1} \qquad [11]$$

where I is the inertia of the rotor, $\omega_{rot}$ is the rotational speed, and t1 and t2 is the start and end time of the observation period,
e.g. one revolution.

**3.4. Performance curves**

15   First the observations are binned based on their wind speed values. In this study bins of 0.5 m/s ranging from 3 to 18m/s are
used. For each bin the mean wind speed is calculated as well as the mean power/flap load, and then the power and flap load
performance curves are generated by linear interpolation between these mean values.

**3.5. Error measure**

Typically the standard deviation of the binned power observations are used to evaluate the uncertainty of one power curve

20   (IEC 61400-12-1, 2005). In this study however, we want to compare several power or load curves. 
[revised manuscript text omitted]